# ECSEL: Explainable Classification via Signomial Equation Learning

**Adia Lumadjeng** [1 2]  **Ilker Birbil** [1]  **Erman Acar** [2 3]

## Abstract

We introduce ECSEL, an explainable classification method that learns formal expressions in the form of signomial equations, motivated by the observation that many symbolic regression benchmarks admit compact signomial structure. ECSEL directly constructs a structural, closed-form expression that serves as both a classifier and an explanation. On standard symbolic regression benchmarks, our method recovers a larger fraction of target equations than competing state-of-the-art approaches while requiring substantially less computation. Leveraging this efficiency, ECSEL achieves classification accuracy competitive with established machine learning models without sacrificing interpretability. Further, we show that ECSEL satisfies some desirable properties regarding global feature behavior, decision-boundary analysis, and local feature attributions. Experiments on benchmark datasets and two real-world case studies (e-commerce and fraud detection) demonstrate that the learned equations expose dataset biases, support counterfactual reasoning, and yield actionable insights.

## 1. Introduction

Modern machine learning often trades explainability for predictive performance. Deep neural networks achieve remarkable accuracy but obscure the reasoning behind predictions. In high-stakes domains such as medicine, finance, and human resources, understanding the underlying decision process is as important as accuracy. Surprisingly, simple, well-chosen formulations can capture the essential structure in data, delivering both accurate predictions and transparent, human-readable explanations (Gerwin, 1974).

[1]Amsterdam Business School, University of Amsterdam, Amsterdam, the Netherlands [2]Institute for Informatics, University of Amsterdam [3]Institute for Logic, Language and Computation, University of Amsterdam. Correspondence to: Adia Lumadjeng <a.c.lumadjeng@uva.nl>, Erman Acar <e.acar@uva.nl>.

*Proceedings of the 43rd International Conference on Machine Learning*, Seoul, South Korea. PMLR 306, 2026. Copyright 2026 by the author(s).

Symbolic Regression (SR) generates inherently interpretable models by producing explicit mathematical formulas that serve as explanations, making it appealing for settings where transparency is essential. This has fueled interest in SR not only as a scientific discovery tool but also as a foundation for interpretable prediction models. Conventional SR approaches, such as Genetic Programming (GP), typically rely on stochastic, population-based search heuristics that are computationally expensive and struggle with high-dimensional datasets.

General-purpose SR methods target arbitrary functional forms, but the equations they are benchmarked against often reflect domain-specific characteristics. The AI Feynman benchmark dataset for SR (Udrescu & Tegmark, 2020) consists of 100 physics equations of which 45 are signomial functions. Figure 1 compares how our method recovers these structurally tractable equations relative to existing approaches.

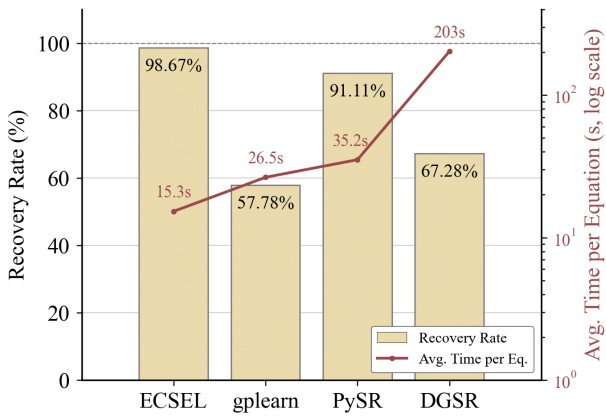

*Figure 1.* Equation recovery rates and average computation time on 45 AI Feynman signomial equations. Comparison of ECSEL (ours), general-purpose SR methods gplearn, PySR, and DGSR (state-of-the-art), averaged over five random seeds (42–46). DGSR timeout cases (> 900s) excluded from time average.

As shown in Figure 1, established SR methods, including the state-of-the art deep learning approach DGSR (Holt et al., 2023), do not fully exploit this simplicity. Baseline methods were evaluated without restricting to signomial forms; however, constraining gplearn (Stephens, 2016) and PySR to such forms did not improve performance, while

DGSR's architecture precluded such constraints. This observation suggests an opportunity: rather than relying on a computationally expensive heuristic search over vast expression spaces, we can directly target this prevalent structural pattern.

Although our formulation is inspired by SR benchmarks, its utility extends far beyond equation discovery. By pairing signomial models with a softmax layer and assigning each class its own signomial, we construct a classifier that delivers competitive performance while producing global and human-readable explanations. The resulting model generates not just predictions but explicit equations that describe how features influence class probabilities. This bridges the gap between the interpretability of SR and the practical demands of modern classification tasks, offering a path toward explainable AI in high-stakes applications.

The primary contributions of this work are:

1. We identify signomial functions as a prevalent structure in symbolic regression benchmarks and propose *ECSEL*, an explainable classifier that exploits this structure. While signomials and gradient-based optimization are individually well-established, our contribution lies in combining them into an inherently interpretable classifier with sparsity-inducing regularization. We show ECSEL achieves competitive classification performance with established benchmark methods while maintaining inherent interpretability through closed-form expressions.

2. We demonstrate that ECSEL recovers target equations in SR benchmarks more frequently than competing state-of-the-art approaches while requiring substantially less computation.

3. We establish that ECSEL satisfies *desirable* analytical properties enabling three levels of explanation: global feature behavior through closed-form elasticities, decision-boundary analysis via exact margin sensitivities, and local feature attributions through log-space decompositions.

4. Through case studies on e-commerce and fraud detection datasets, we demonstrate that ECSEL's learned equations expose dataset characteristics and yield actionable insights.

Code for reproducing all experiments is available at github.com/AdiaLumadjeng/ecsel.

## 2. Related Work

Unlike traditional regression, which fits parameters to a predefined equation, Symbolic Regression (SR) searches for both model structure and parameters. The primary output is a concise, human-interpretable equation, making SR powerful for scientific discovery and explainable AI. However, the combinatorial search makes the problem NP-hard (Virgolin & Pissis, 2022).

Heuristic approaches navigate the search space using evolutionary algorithms. Genetic Programming (GP) iteratively evolves candidate expressions through selection, crossover, and mutation (Koza, 1992). Modern tools like PySR remain competitive through simulated annealing and efficient constant optimization (Cranmer, 2023). Udrescu & Tegmark (2020) introduced the AI Feynman algorithm, using neural networks to detect physical properties like symmetry and separability as heuristics to recursively decompose problems. The authors developed the AI Feynman benchmark, now a standard for SR evaluation. Jin et al. (2019) use MCMC sampling to incorporate priors and estimate uncertainty. These methods are powerful general-purpose tools, but do not exploit structural patterns that may be prevalent in specific domains.

Deep learning approaches reframe SR as sequence generation or representation learning. Deep Symbolic Regression (DSR) uses an RNN to generate expressions token-by-token with risk-seeking policy gradients (Petersen et al., 2021). Current state-of-the-art leverages Transformers: NeSymRes pre-trains on millions of synthetic equations to predict formulas in a single forward pass (Biggio et al., 2021), while DGSR employs permutation-invariant encoders for superior generalization (Holt et al., 2023). Hybrid methods combine deep learning with heuristic search: neural-guided GP uses RNNs to initialize evolutionary algorithms (Mundhenk et al., 2021b). Kolmogorov-Arnold Networks (KANs) offer an interpretable alternative to MLPs using learnable spline functions simplifiable into symbolic formulas (Liu et al., 2025). While these methods push the state of the art on general benchmarks, none are designed to exploit specific function classes such as signomials efficiently.

Signomial functions appear frequently in SR benchmarks, but have not yet been studied as learning models in their own right. First introduced by Duffin & Peterson (1973) in the context of geometric programming, signomials there serve as objectives to be minimized rather than models to be learned. Our setting is fundamentally different: we treat signomials as a model class and minimize prediction error against observed labels, bringing signomials into the supervised learning framework.

Explainable classification has become a central theme in modern machine learning, driven by the need to make transparent and trustworthy decisions. The field develops inherently interpretable models, such as decision trees, which offer direct, human-readable logic. Generalized Additive Models (GAMs) (Lou et al., 2013) extend this by learn-

ing nonlinear shape functions per feature while preserving additivity, with recent work incorporating neural networks (Agarwal et al., 2021). Complementary approaches use sparsity-inducing regularization (Louizos et al., 2018) to select interpretable feature subsets, balancing expressiveness with transparency.

More recent efforts have extended interpretability to complex black-box classifiers through post-hoc explanation techniques such as LIME (Ribeiro et al., 2016), SHAP (Lundberg & Lee, 2017), and Integrated Gradients (Sundararajan et al., 2017). Despite their popularity, critics argue that post-hoc explanations can be unreliable or misleading for high-stakes decisions, advocating instead for inherently interpretable architectures (Rudin, 2018). ECSEL follows this tradition, learning a closed-form signomial expression that serves as both classifier and explanation.

## 3. Approach

We begin with signomial functions and establish their universal approximation property (Section 3.1). We formulate ECSEL, our signomial-based classifier, and derive a set of desirable properties that enable interpretation regarding global feature behavior, decision-boundaries, and local feature attribution (Section 3.2).

### 3.1. Signomial Functions

Let $\{(x_i, y_i)\}_{i=1}^{N}$ denote a dataset, where $x_i = (x_{i1}, \ldots, x_{im}) \in \mathbb{R}^m$ is the feature vector for sample $i$ and $y_i$ is the corresponding target. We consider signomial functions, that is, a finite sums of power-law terms, with real-valued constants and real-valued exponents.

A signomial with $K$ terms is defined as

$$z(x_i) = \sum_{k=1}^{K} \alpha_k \prod_{j=1}^{m} x_{ij}^{\beta_{k,j}}, \tag{1}$$

where $\alpha_k \in \mathbb{R}$ are coefficients and $\beta_{k,j} \in \mathbb{R}$ are exponents.

Despite their simplicity, signomials are highly expressive. A single-term signomial ($K = 1$) naturally represents power laws, inverse relationships ($\beta_j < 0$), and root-type effects ($\beta_j = \frac{p}{q}$), functional forms that appear frequently in scientific equations. Empirically, 45 out of 100 equations in the AI Feynman benchmark are signomials.

Theoretically, signomials form a rich function class. We establish that signomials are dense in the space of continuous functions on compact subsets of the positive orthant, meaning they can approximate any continuous function to arbitrary precision. The following result places signomials alongside neural networks as universal approximators, while being tailored to multiplicative power-law relationships.

**Theorem 3.1** (Universal Approximation for Signomials). *Let $D \subset \mathbb{R}_{>0}^m$ be compact. For any $f \in C(D, \mathbb{R})$ and $\epsilon > 0$, there exists a signomial $z(x) = \sum_{k=1}^{K} \alpha_k \prod_{j=1}^{m} x_j^{\beta_{k,j}}$ such that $\sup_{x \in D} |f(x) - z(x)| < \epsilon$.*

*Proof sketch.* Via logarithmic transformation, signomials map to exponentials of linear functions, establishing a homeomorphism between $C(D, \mathbb{R})$ on $D \subset \mathbb{R}_{>0}^m$ and continuous functions on the log-transformed domain. This enables application of the Stone-Weierstrass theorem (Stone, 1948). Full proof can be found in Appendix A.

**From structure to interpretability.** Importantly, the same structural properties that make signomials compact also enable interpretability. For a single-term signomial, each exponent $\beta_{k,j}$ directly encodes how a feature scales model output, with its sign and magnitude characterizing the direction and strength of the feature's effect. We leverage this direct relationship between model parameters and feature contribution to develop the analytical properties presented in Section 3.2.

**Faithfulness of exponent magnitude.** A natural question is whether exponent magnitude truly reflects feature contribution. For the single-term case ($K = 1$), we can formally establish this relationship: when a feature $x_k$ is scaled by any factor $r$, the signomial output changes by $r^{\beta_k}$. This means features with larger $|r^{\beta_k}|$ induce proportionally larger output responses under the same perturbation, creating a faithful ordering where $|\beta_k| > |\beta_\ell|$ implies feature $k$ contributes more strongly than feature $\ell$. While this result is shown for $K = 1$ (see Proposition B.1 in Appendix B), it provides intuition for why exponent-based interpretation works in the muti-term setting: each term's exponents capture how feature scale that term's contribution to the overall prediction.

**Classification with ECSEL.** *Explainable Classification via Signomial Equation Learning* (ECSEL) formulates classification by learning class-specific signomial score functions (see Figure 2). For a multi-class problem with $C$ classes, the score for class $c$ is defined as

$$z_c(x) = \sum_{k=1}^{K} \underbrace{\alpha_{c,k} \prod_{j=1}^{m} x_j^{\beta_{c,k,j}}}_{z_{c,k}(x)}, \tag{2}$$

with learnable parameters $\alpha_{c,k} \in \mathbb{R}$ and $\beta_{c,k,j} \in \mathbb{R}$ for each class $c \in \{0, \ldots, C-1\}$, component $k \in \{1, \ldots, K\}$, and feature $j \in \{1, \ldots, m\}$. Since real-valued exponents require positive arguments, features are preprocessed via affine transformations to ensure $x_{ij} > 0$ for all $j$ (e.g., scaling to $[1, 10]$). The number of terms $K$ controls model complexity: $K = 1$ yields a single power-law score per class, while larger $K$ allows additive composition of multiple interacting components.

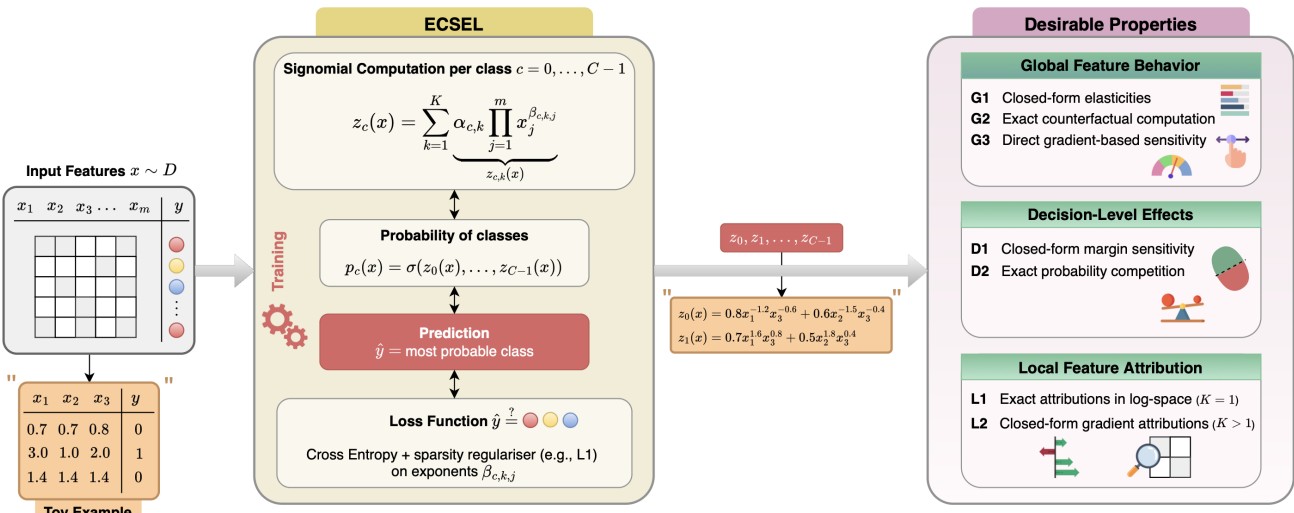

*Figure 2.* Overview of ECSEL's computation flow and analytical properties. Input features are transformed through class-specific signomial functions to produce score functions $z_c(x)$, which are then converted to class probabilities via softmax. The signomial structure enables three categories of desirable properties: global feature behavior (elasticities, counterfactuals, sensitivity), decision-level effects (margin analysis, probability competition), and local feature attributions (exact for $K = 1$, gradient-based for $K > 1$).

**Training objective and regularization.** We train the model by minimizing the cross-entropy loss with sparsity inducing regularization on the exponents:

$$\mathcal{L}(\alpha, \beta) = -\frac{1}{N} \sum_{i=1}^{N} \log p_{y_i}(x_i) + \lambda \sum_{c,k,j} |\beta_{c,k,j}|, \quad (3)$$

where $y_i \in \{0, \ldots, C-1\}$ is the class label for sample $i$, $p_{y_i}(x_i)$ is the predicted probability for the true class, $N$ is the number of samples, and $\lambda$ controls controls the regularization strength applied to all exponents.

We use $\ell_1$ regularization to encourange automatic feature selection by driving irrelevant exponents toward zero, producing more interpretable equations with fewer active features per component. We remark that $\ell_1$ is not essential for our framework and other regularizers can also be used. We use gradient-based optimization methods and apply numerical stability techniques (e.g., feature scaling, logarithmic transformation) to handle the power-law terms effectively.

**Probability transformation.** Predicted probabilities are obtained using softmax over class scores or, in the binary case, a sigmoid applied to a single score or a two-class softmax. These formulations differ in parameterization and optimization dynamics, which can lead to different predictions. The resulting scores are not guaranteed to be calibrated; post-hoc techniques such as temperature scaling (Guo et al., 2017) could be applied if needed.

ECSEL can be viewed as a per-class network with exponential activations on log-transformed features; while architecturally related to shallow MLPs, the signomial parameterization enables the closed-form interpretability properties de-

rived in Section 3.2. A detailed comparison against matched per-class MLPs is provided in Appendix E.5.

### 3.2. Desirable Properties of ECSEL

Beyond the inherent interpretability of the learned equations, we reveal several properties induced by the functional form of ECSEL that enable interpretation of its predictions. These properties explain how features influence class scores, decision boundaries, and individual predictions. We group them into global feature behavior, decision-level effects, and local feature attributions.

**Global feature behavior.** We distinguish between two related quantities. The *elasticity* (log-log derivative) of a class score with respect to a feature is

$$E_{c,j}(x) := \frac{\partial \log z_c(x)}{\partial \log x_j}, \quad (4)$$

which measures proportional sensitivity of the score. The *log-gradient* of the score is

$$G_{c,j}(x) := \frac{\partial z_c(x)}{\partial \log x_j}, \quad (5)$$

which measures absolute sensitivity under proportional feature changes. When $z_c(x) > 0$, the two are related by $G_{c,j}(x) = z_c(x) E_{c,j}(x)$.

We first describe properties that characterize how individual features influence class scores across the input space.

PROPERTY G1 (GLOBAL FEATURE ATTRIBUTION). The signomial structure enables *direct* computation of global fea-

ture importance through gradient-based attribution. Specifically, the elasticity $E_{c,j}(x)$ as defined in Eq. (4), admits a closed-form expression in terms of the model parameters. For ECSEL scores $z_c(x) = \sum_{k=1}^{K} z_{c,k}(x)$, and for inputs such that $z_c(x) > 0$, we have

$$E_{c,j}(x) = \sum_{k=1}^{K} \frac{z_{c,k}(x)}{z_c(x)} \beta_{c,k,j}. \tag{6}$$

For $K = 1$, this reduces to the constant elasticity $E_{c,j}(x) = \beta_{c,j}$, whereas for $K > 1$, it varies with $x$ through the component responsibilities $\frac{z_{c,k}(x)}{z_c(x)}$.

PROPERTY G2 (COUNTERFACTUAL REASONING). The signomial structure enables *exact* computation of counterfactual effects under proportional feature perturbations. Specifically, the effect of scaling a feature $x_j$ by a factor $q > 0$ induces an analytic score transformation $z_c(x) \mapsto z_c^{\text{new}}(x)$, which can be computed directly from the original score $z_c(x)$ and model parameters. This allows direct assessment of "what-if" scenarios without retraining or re-evaluation. For ECSEL, this reduces to

$$z_c^{\text{new}}(x) = \sum_{k=1}^{K} q^{\beta_{c,k,j}} z_{c,k}(x). \tag{7}$$

PROPERTY G3 (GRADIENT-BASED SENSITIVITY). For small proportional changes to features, class scores respond linearly to first order. Specifically, scaling feature $x_j$ by a small factor $(1 + \varepsilon)$ induces a score change

$$z_c(x) \mapsto z_c(x) + \varepsilon \cdot \frac{\partial z_c(x)}{\partial \log x_j} + \mathcal{O}(\varepsilon^2), \tag{8}$$

enabling tractable sensitivity analysis through the score log-gradient $G_{c,j}$ defined in Eq. (5). For ECSEL, this log-gradient admits the closed form $G_{c,j}(x) = \sum_{k=1}^{K} \beta_{c,k,j} z_{c,k}(x)$, yielding $z_c(x) \mapsto z_c(x) + \varepsilon G_{c,j}(x) + \mathcal{O}(\varepsilon^2)$. For single-component models ($K = 1$), the response simplifies to $z_c(x) \mapsto z_c(x)(1 + \varepsilon \beta_{c,j})$, showing that the score scales proportionally with the exponent.

**Decision-level effects.** Next, we consider properties that describe how features affect class competition and decision boundaries.

PROPERTY D1 (DECISION BOUNDARY SENSITIVITY). The influence of a feature on the decision boundary between classes $c$ and $c'$ can be characterized by the sensitivity of the score margin to proportional feature changes,

$$\frac{\partial\big(z_c(x) - z_{c'}(x)\big)}{\partial \log x_j}. \tag{9}$$

For ECSEL, this reduces to a closed-form expression, which when $z_c(x), z_{c'}(x) > 0$, can be written as

$$z_c(x) E_{c,j}(x) - z_{c'}(x) E_{c',j}(x). \tag{10}$$

When $K = 1$, the expression simplifies to $z_c(x)\beta_{c,j} - z_{c'}(x)\beta_{c',j}$, directly revealing how exponent differences drive class competition.

PROPERTY D2 (DIRECT PROBABILITY COMPETITION). Feature effects on predicted class probabilities reflect competitive interactions between classes through normalization. The sensitivity of a class probability $p_c(x)$ to proportional feature changes, $\frac{\partial p_c(x)}{\partial \log x_j}$, can be computed exactly from the signomial structure's class-specific sensitivities and predicted probabilities. For ECSEL with softmax-normalized scores, and using Eq. (5), this sensitivity admits the closed form

$$\frac{\partial p_c(x)}{\partial \log x_j} = p_c(x)\big(G_{c,j}(x) - \sum_r p_r(x) G_{r,j}(x)\big). \tag{11}$$

A feature increases class $c$ probability only when its log-gradient exceeds the probability-weighted average across all classes.

**Local feature attribution.** Finally, we derive properties that characterize how individual features contribute to a specific prediction, enabling instance-level explanations.

PROPERTY L1 (EXACT LOG-SPACE ADDITIVITY). The individual component of a signomial function becomes additive in log-space. Specifically, for components $z_{c,k}(x) > 0$, the log-score can be expressed as a baseline term plus a sum of feature-specific contributions. For ECSEL, taking logarithms of each signomial component yields:

$$\log z_{c,k}(x) = \log z_{c,k}(b) + \sum_j \beta_{c,k,j} \log \frac{x_j}{b_j}, \tag{12}$$

where $b$ is a reference baseline (e.g., geometric mean). For $K = 1$, this provides exact log-space additivity for the entire class score, similar to the well-known SHAP values (Lundberg & Lee, 2017). However, for $K > 1$ the aggregated score leads to an approximation as we give next.

PROPERTY L2 (GRADIENT-BASED LOCAL ATTRIBUTIONS). When exact additive decomposition is not available (i.e., for aggregated nonlinear functions), local feature attributions can be obtained via first-order linearization around a reference input. For ECSEL with $K > 1$, the aggregated score (defined in Eq. (2)) is nonlinear, so the contribution of feature $x_j$ is approximated by

$$\phi_j^{(z_c)}(x) \approx G_{c,j}(x^*)\big(\log x_j - \log x_j^*\big), \tag{13}$$

where $G_{c,j}(x^*)$ is the score log-gradient in Eq. (5) evaluated at baseline $x^*$. An analogous expression holds for class probabilities using the probability gradient from Property D2. Overall, we obtain gradient-based local attributions through closed-form gradients rather than expensive sampling procedures.

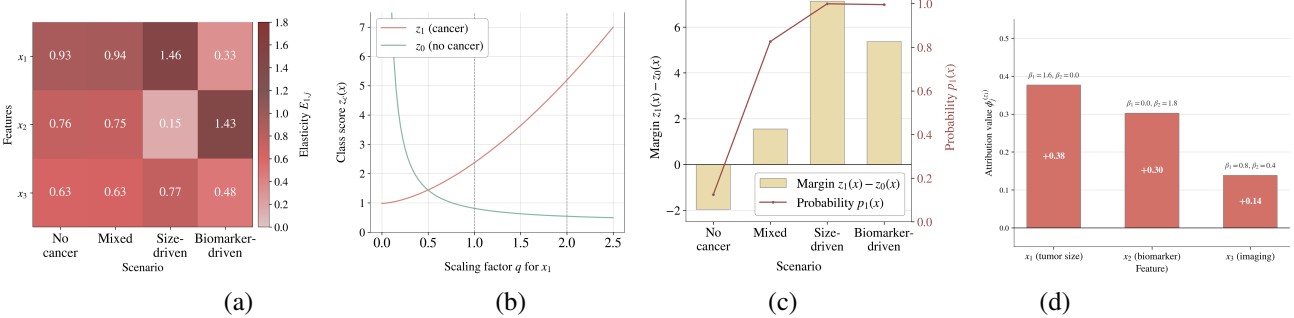

(a)         (b)         (c)         (d)

*Figure 3.* **Interpretability properties illustrated on a toy cancer screening example** with tumor size ($x_1$), biomarker level ($x_2$), and imaging irregularity ($x_3$). Four scenarios: no-cancer, mixed, size-driven and biomarker-driven. **(a)** Feature elasticities show context-dependent importance (Property G1). **(b)** Exact counterfactual under scaling tumor size by $q$ (Property G2). **(c)** Decision margin $M_{1,0}$ and cancer probability $p_1$ across scenarios (Properties D1–D2). **(d)** Gradient-based attributions $\phi_j \approx G_{c,j}(x^*)(\log x_j - \log x_j^*)$ to the cancer class for mixed sample at baseline $x^* = (1, 1, 1)$ (Property L2).

Properties L2 and G1 are complementary: G1 characterizes local sensitivity at any point, and L2 turns this into an additive feature contribution relative to a baseline by evaluating the same log-gradient $G_{c,j}$ at a reference point $x^*$.

**Theorem 3.2** (Interpretability properties of ECSEL). *Let $\{(x_i, y_i)\}_{i=1}^N$ be a dataset with $x_i \in \mathbb{R}_{>0}^m$ denoting the pre-processed positive feature representation (e.g., via affine shifting and rescaling as described in Section 3.1). Let ECSEL learn class scores as defined in Eq. (2) with component scores $z_{c,k}(x) = \alpha_{c,k} \prod_{j=1}^m x_j^{\beta_{c,k,j}}$ and log-gradients $G_{c,j}(x) = \sum_{k=1}^K \beta_{c,k,j} z_{c,k}(x)$. Then, for all $x \in \mathbb{R}_{>0}^m$ with $z_c(x) > 0$ where required, the resulting classifier satisfies Properties G1–G3, D1–D2, and L1–L2.*

*Proof sketch.* The results follow from ECSEL's signomial structure through closed-form differentiation. Properties G1–G3 use power-law derivatives and Taylor expansions; D1–D2 apply these to margins and softmax; L1 exploits log-linearity of components; L2 uses local linearization for aggregated functions. See Appendix C for the full proof.

**Illustrative example.** Figure 3 demonstrates ECSEL's properties on a synthetic cancer screening example with features $x_1$ (tumor size), $x_2$ (biomarker level), $x_3$ (imaging irregularity). Let each class have two signomial components capturing distinct diagnostic pathways: the no-cancer score $z_0 = 0.8x_1^{-1.2}x_3^{-0.6} + 0.6x_2^{-1.5}x_3^{-0.4}$ decreases with tumor size and biomarker level, while the cancer score $z_1 = 0.7x_1^{1.6}x_3^{0.8} + 0.5x_2^{1.8}x_3^{0.4}$ increases with both. We consider four patient profiles: no-cancer $(0.7, 0.7, 0.8)$, mixed $(1.4, 1.4, 1.2)$, size-driven $(3.0, 1.0, 2.0)$, and biomarker-driven $(1.0, 3.0, 2.0)$. Panel (a) shows context-dependent feature importance: $x_1$ dominates in the size-driven case, $x_2$ in the biomarker-driven case. Panel (b) shows scaling $x_1$ drives the cancer score up while reducing the no-cancer score. Panel (c) shows the margin turning positive only for the size- and biomarker-driven scenarios. Panel (d) shows

all features contributing positively to the cancer score for the mixed scenario, with $x_1$ contributing most.

Several of the properties presented above have direct counterparts in standard explanation methods: G1 corresponds to global SHAP attributions, G3 to LIME's local surrogate, and L1, L2 to additive SHAP decompositions. A key distinction is that ECSEL's versions are derived analytically from the model parameters rather than estimated via sampling, and are exact for $K = 1$.

## 4. Symbolic Regression Benchmark

Although ECSEL is designed for classification, its functional form frequently appears in symbolic regression datasets. We show that ECSEL recovers a larger fraction of target signomial equations than state-of-the-art neural symbolic regression methods while doing so with substantially lower computation time.

### 4.1. Experimental Setup

For symbolic regression tasks, we adapt ECSEL by replacing the cross-entropy with Mean Squared Error (MSE), learning a continuous-valued signomial function $z(x)$ to fit the target equation:

$$\mathcal{L}_{\text{SR}}(\alpha, \beta) = \frac{1}{N} \sum_{i=1}^N (y_i - z(x_i))^2 + \lambda \sum_{k,j} |\beta_{k,j}|. \quad (14)$$

We retain $\ell_1$ regularization on exponents to encourage compact, interpretable equations, consistent with the symbolic regression goal of recovering simple expressions rather than maximizing predictive accuracy alone.

To navigate the nonconvex landscape, we use multi-start staged optimization strategy: for single-term signomials ($K = 1$), we apply L-BFGS-B, which is well-suited to

the low-dimensional and smooth objective. For multi-term signomials ($K > 1$), the higher-dimensional and nonconvex parameter space benefits from adaptive stochastic optimization, and so we adopt a staged strategy: Adam-based structure discovery with stronger sparsity regularization, followed by refinement with reduced regularization, and a final L-BFGS polishing step initialized from the best Adam solution.

**Baselines and benchmarks.** We compare ECSEL against three state-of-the-art neural symbolic regression methods: Deep Generative Symbolic Regression (DGSR) (Holt et al., 2023), Neural-Guided Genetic Programming (NGGP) (Mundhenk et al., 2021b), and NeSymRes (Biggio et al., 2021). These methods are evaluated on a collection of signomial-rich benchmarks, including the *AI Feynman* (Udrescu & Tegmark, 2020), *Livermore* (Mundhenk et al., 2021a), *Jin* (Jin et al., 2019), and *Korns* (Korns, 2013) problem sets, as well as equations from a collection of synthetic power-law expressions compiled by the authors of DGSR (Holt et al., 2023).

Following standard practice, all experiments use Gaussian noise ($\sigma = 0.01$) and are conducted using the DGSR authors' publicly available benchmarking suite[1] with a 15-minute time limit per equation. Full technical specifications and baseline configurations are provided in Appendix D.

### 4.2. Results

We evaluate performance using the *symbolic recovery rate*, defined as the fraction of five random seeds (42–46) recovering the target expression up to algebraic equivalence, and average solving time. Detailed per-equation performance is presented in Tables 3 (Appendix D).

ECSEL achieves a global average recovery rate of 95.86%, substantially outperforming DGSR (59.10%), NGGP (58.54%), and NeSymRes (56%), while also requiring significantly less computation time. ECSEL is also significantly faster, averaging 86.4 seconds per equation, compared to 612.9, 468.7, and 126.3 seconds for DGSR, NGGP, and NeSymRes, respectively; several DGSR and NGGP runs fail to recover an expression within the time limit.

The runtime variance in Table 3 reflects structural complexity: equations from *Top* through *Constant-6* consist of a single signomial term ($K = 1$) and are solved rapidly, while subsequent equations contain multiple terms ($K > 1$) involving a higher-dimensional and more nonconvex optimization landscape. Performance gains are particularly pronounced for expressions involving rational powers and inverse-polynomial structure, where several competing methods frequently fail or time out. Notably, even when

---

[1] https://github.com/samholt/DeepGenerativeSymbolicRegression

exact symbolic recovery is unsuccessful, ECSEL typically yields near-perfect numerical approximations ($R^2 \approx 1$). Table 4 in Appendix D.4 shows the five equations generated by ECSEL with a recovery rate of less than 100%.

## 5. Classification Benchmark

This benchmark evaluates the classification performance of ECSEL against five established machine learning methods, representing linear, tree-based, kernel, and neural approaches. We assess whether the interpretable power-law structure of signomials can match the predictive accuracy of both simple linear baselines and complex black-box ensembles.

### 5.1. Experimental Setup

For classification, ECSEL utilizes the Adam optimizer with gradient clipping to ensure numerical stability of the exponents. To ensure robust performance estimates, we employ 5-fold stratified cross-validation. For each model, hyperparameters are optimized over 30 trials using Optuna's TPE sampler. We apply early stopping based on an internal validation split to prevent overfitting and ensure the model generalizes effectively. The best-performing configurations are then retrained on the full training set and evaluated on a 20% held-out test set.

**Baselines and benchmarks.** We compare ECSEL against Logistic Regression (LR), Random Forest (RF), XGBoost, Support Vector Machines (SVM), and a Multi-Layer Perceptron (MLP). These methods are evaluated on 11 standard binary and multi-class benchmarks, across domains such as medical diagnosis, financial risk, and criminal justice. The suite includes datasets of varying scales and difficulty, ranging from 150 to nearly 400,000 samples. All input features are scaled to [1,10] using MinMax scaling to ensure the positivity required by the signomial structure and to maintain consistent preprocessing across all baselines. Detailed dataset characteristics and hyperparameter search spaces are provided in Appendices E.1 and E.3.

### 5.2. Results

We report accuracy, F1-score, and minority class recall to account for class imbalances. Table 1 presents representative results on three datasets; complete results across all methods and datasets are in Table 10 (Appendix E.4).

ECSEL achieves the highest F1-score on 4 of 11 datasets (SEEDS, HEARTS, ILPD, and COMPAS), demonstrating that its constrained functional form remains highly competitive. Table 1 illustrates this pattern: ECSEL substantially outperforms baselines on ILPD (+11.36% F1 over XGBoost with a 36-point gain in minority recall) and narrowly leads on COMPAS. On datasets where ECSEL is not the top per-

*Table 1.* Performance comparison of ECSEL against strongest baselines on selected datasets.

| Dataset | Method | Acc. | F1 | Min. Recall |
|---|---|---|---|---|
| ILPD | LR | 71.55 | 58.45 | 3.03 |
| | XGBoost | 72.41 | 63.03 | 6.06 |
| | ECSEL | **75.86** | **74.39** | **42.42** |
| COMPAS | RF | 67.69 | 67.63 | 60.40 |
| | XGBoost | 68.18 | 68.08 | 62.54 |
| | ECSEL | **68.47** | **68.36** | **62.82** |
| TRANSFUSION | RF | 78.66 | 77.40 | 36.11 |
| | XGBoost | **80.06** | **78.72** | 38.89 |
| | ECSEL | 79.33 | 77.95 | **41.67** |

former, such as TRANSFUSION, it typically trails the best baseline by less than one percentage point while often maintaining superior minority recall (41.67% vs. 38.89% on TRANSFUSION). Overall, ECSEL ranks within one percentage point of the best method on 9 of 11 datasets. On the two remaining datasets with extreme class imbalance (SKINNONSKIN, MAMMOGRAPHY), ensemble methods retain an advantage, though ECSEL maintains competitive minority recall. While ECSEL's training times are higher than some baselines, it shifts cost from explanation to training: once trained, all attributions are obtained in closed form at negligible cost, whereas black-box models require additional post-hoc procedures such as SHAP or LIME.

A per-dataset head-to-head comparison with standard deviations over 5-fold cross-validation for F1-score is provided in Table 9; complete results across all metrics and methods are in Table 10. A dedicated interpretability analysis, including comparisons with SHAP and LIME, is provided alongside the Online Shopping Intention case study of Section 6.1.

## 6. Case Studies

In this section, we demonstrate ECSEL across two classification domains with distinct characteristics: online purchase intent prediction and large-scale financial fraud detection.

### 6.1. Case Study: Online Shopping Intention

We apply ECSEL to predict e-commerce purchase intent using the Online Shoppers Intention dataset (Sakar et al., 2018), which contains 12,330 user sessions with 15.5% purchase rate. ECSEL learns a seven-feature signomial that captures the key drivers of purchase conversion:

$$z = 0.10 \frac{PageValues^{0.47} \, Month^{0.07} \, PVER^{1.09} \, ShopIntensity^{0.66}}{ExitRates^{0.41} \, Administrative^{0.14} \, IsReturn^{0.04}} \quad (15)$$

Predicted purchase probabilities are obtained as $p(x) = \sigma(z(x))$, with a decision threshold of $p = 0.559$ selected on the validation set to optimize F1 score. All dataset features can be found in Appendix F.1.

The learned elasticities identify *PageValue_per_ExitRate*

(*PVER*) as the dominant predictor ($\beta = 1.09$), indicating that high-value pages with low exit rates signal purchase intent. *ShopIntensity* and *PageValues* contribute positively, whereas *ExitRates* and *Administrative* exhibit negative elasticities, reflecting disengagement and help-seeking behavior. Consistent with the dataset's class imbalance, the default prediction corresponds to no purchase, requiring sufficiently strong positive signals to cross the decision boundary.

When comparing ECSEL against efficient baseline methods (see Table 14 for the results), ECSEL does not achieve the highest accuracy or F1 score, but its performance remain competitive. On this imbalanced dataset, minority class recall is particularly important: ECSEL achieves 75.7% recall, on par with MLP (75.4%) and SVM (76.6%), and substantially exceeding LR (67.7%), RF (70.3%) and XGBoost (66.4%), all while providing full interpretability through its closed-form expression and training in just 5.5 seconds.

**Comparison with explanation methods.** To validate the connections between ECSEL's analytical properties and standard explanation methods, we compute feature attributions for ECSEL via its G1 property alongside TreeSHAP for RF and XGBoost, KernelSHAP for MLP, LinearSHAP for LR, and LIME for all methods. Table 2 summarizes the explanation methods, top-3 features, and computation times over the full test set. All nonlinear methods identify *PVER* as the dominant predictor, while LR assigns highest importance to *Month*, likely reflecting its limited capacity to capture multiplicative feature interactions. ECSEL produces attributions at zero computational cost versus up to 28.5s for KernelSHAP. ECSEL's global feature rankings align closely with tree-based methods (Spearman $\rho \geq 0.80$, $p < 0.001$) and show weaker, non-significant agreement with MLP; full pairwise correlation results are reported in Table 16 (Appendix F.5).

*Table 2.* Explanation methods, top-3 features, and computation time on the OSI test set. Feature abbreviations: *PVER* (*PageValue_per_ExitRate*), *SI* (*ShopIntensity*), *PV* (PageValues), *Mo* (*Month*), *IR* (*IsReturning*), *Ad* (*Administrative*), *PR* (*ProductRelated*), *ER* (*ExitRates*).

| Method | Explainer | Time (s) | Top-3 features |
|---|---|---|---|
| ECSEL | Exact exponents | 0.1 | *PVER, SI, PV* |
| LR | Exact coefficients | 0.1 | *Mo, IR, Ad* |
| | LinearSHAP | 0.1 | *PVER, Mo, PR* |
| | LIME | 5.3 | *PVER, Mo, PR* |
| RF | TreeSHAP | 1.5 | *PVER, PV, SI* |
| | LIME | 32.0 | *PVER, PV, ER* |
| XGBoost | TreeSHAP | 0.1 | *PVER, Mo, SI* |
| | LIME | 7.7 | *PVER, PR, ER* |
| MLP | KernelSHAP | 28.5 | *PVER, PR, Mo* |
| | LIME | 5.7 | *PVER, PR, PV* |

Complete experimental details, regularization trade-off analysis, and visual demonstrations of ECSEL's desirable properties are provided in Appendix F. Appendix F.5 additionally includes supporting results for the comparison with explanation methods above, including a direct G3 vs. LIME comparison, motivated by their shared role as local attribution methods.

### 6.2. Case Study: PaySim Fraud Detection

The PaySim dataset (Lopez-Rojas et al., 2016) is a synthetic mobile money transaction log with approximately 6.3 million transactions over 30 days and severe class imbalance (0.13% fraud rate). We compare ECSEL's performance against Deep Symbolic Classification (DSC) (Visbeek et al., 2024), which previously applied symbolic regression to this dataset. Full experimental details are provided in Appendix G.

The signomial learned by ECSEL is given by:

$$z = -0.07 \frac{A^{0.02}\, P^{0.03}}{exO^{0.03}\, exD^{0.16}\, CO^{0.14}\, T^{0.06}\, D^{0.03}}$$
$$+ 0.09 \frac{OBO^{1.42}}{NBO^{0.04}\, exD^{0.07}\, CO^{0.06}\, D^{0.06}\, P^{0.06}}. \quad (16)$$

Here $A$ is the transaction *amount*. Indicators *exO* and exD denote whether the origin or destination accounts are external (*externalOrig*, *externalDest*), *OBO* and *NBO* are the origin account balances before and after transaction (*oldbalanceOrig* and *newbalanceOrig*). The transaction types *Transfer*, *Debit*, *Payment* and *CashOut* are denoted by *T, D, P* and *CO* respectively. All features can be found in Table 18 (Appendix G).

ECSEL achieves an F1 score of 79.08%, with recall of 68.10% and precision of 94.27% at the optimal threshold of 0.904. The higher threshold is expected in heavily imbalanced settings, where the model must assign high confidence before flagging a transaction as fraud. The selected model uses an $\ell_1$ regularization strength of $\lambda = 2 \times 10^4$, and training on the full dataset requires approximately 16 minutes. For comparison, Visbeek et al. (2024) report an F1 score of 78.0% using DSC on the same dataset, though methodological differences in temporal feature engineering affect direct comparability (see Appendix G.4).

When it comes to the interpretation, the learned signomial in Eq. (16) consists of two terms with distinct roles. The first term (coefficient $-0.07$) acts as a transaction type filter, with *typePayment* in the numerators and fraud-associated types (*typeCashOut* and *typeTransfer*) in the denominator. Since this term carries a negative coefficient, high values reduce the fraud score, effectively filtering out legitimate payment transactions while allowing cash-out and transfer transactions to be flagged.

The second term (coefficient $+0.09$) captures the primary fraud signal through a superlinear relationship with the originator account balance ($\beta = 1.42$). This superlinear scaling reveals that fraudsters disproportionately target high-value accounts, as larger payoffs justify additional effort and risk. The weak negative exponent on *newbalanceOrig* (0.04) suggests lower post-transaction balances slightly increase fraud probability, capturing account draining behavior, although this effect is minimal compared to the dominant balance-targeting pattern.

## 7. Limitations and Future Work

Like other parametric models, ECSEL requires specifying the number of terms $K$ a priori. While this represents a genuine limitation for symbolic regression, it becomes a standard hyperparameter in classification settings.

For high-degree univariate polynomials (e.g., in Nguyen dataset (Nguyen et al., 2011)), other methods can outperform ECSEL in exact symbolic recovery, while ECSEL still converges rapidly to low NMSE and high $R^2$ (see Table 5 in Appendix D).

Future work could explore several promising directions: adaptive structure selection methods to automatically determine $K$; formal characterization of faithfulness guarantees under diverse data distributions; investigation of alternative sparsity-inducing regularizers and their effect on feature selection stability, particularly with correlated features; extensions to regression tasks; and applications to additional high-stakes domains.

## 8. Conclusion

We introduced ECSEL, an explainable classification method that learns interpretable signomial equations as both predictive models and transparent explanations. Our symbolic regression experiments confirm ECSEL's effectiveness at exploiting signomial structure, substantially outperforming state-of-the-art methods in both recovery rate and computational efficiency. This computational efficiency extends to classification tasks, where ECSEL achieves competitive accuracy with black-box baselines. Importantly, ECSEL satisfies formal interpretability properties enabling analysis of global feature behavior, decision-boundaries, and instance-level explanations. Our case studies on e-commerce and fraud detection show practical utility beyond benchmark metrics: ECSEL's formulas reveal clear decision drivers and interpretable patterns invisible to ensemble methods, such as the superlinear wealth-targeting mechanism in fraud detection. By combining computational efficiency with transparent decision-making, ECSEL offers practitioners an actionable alternative to black-box models in high-stakes domains.

## Acknowledgements

The authors thank the reviewers for their constructive feedback. This research was generously supported by the AI4Fintech Initiative at the University of Amsterdam.

## Impact Statement

This paper presents work whose goal is to advance the field of Machine Learning. There are many potential societal consequences of our work, none which we feel must be specifically highlighted here.

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

## A. Universal Approximation Property of Signomial Functions

We show that signomial functions possess the universal approximation property on compact subsets of the positive orthant. Specifically, we prove that the set of all signomials $\mathcal{S}$ is dense in $C(D, \mathbb{R})$, the space of continuous real-valued functions on a compact set $D \subset \mathbb{R}^n_{>0}$, equipped with the uniform norm $\|\cdot\|_\infty$. The key to our proof is a logarithmic transformation which establishes a homeomorphism. This transformation converts signomials into exponentials of linear functions, allowing us to apply the classical Stone-Weierstrass (Stone, 1948) theorem from approximation theory. We restate the theorem from Section 3.1 below.

**Theorem 3.1** (Universal Approximation for Signomials). *Let $D \subset \mathbb{R}^n_{>0}$ be a compact subset of the positive orthant. Then the set of signomials*

$$\mathcal{S} = \left\{ S : S(x_1, \ldots, x_n) = \sum_{k=1}^{K} \alpha_k \prod_{j=1}^{n} x_j^{\beta_{kj}}, \quad K \in \mathbb{N}, \ \alpha_k \in \mathbb{R}, \ \beta_{kj} \in \mathbb{R} \right\}$$

*is dense in $C(D, \mathbb{R})$. That is, for any continuous function $f : D \to \mathbb{R}$ and any $\epsilon > 0$, there exists a signomial $S \in \mathcal{S}$ such that*

$$\sup_{(x_1, \ldots, x_n) \in D} |f(x_1, \ldots, x_n) - S(x_1, \ldots, x_n)| < \epsilon.$$

*Remark* A.1. This theorem provides theoretical justification for using signomial classifiers in machine learning applications where features are naturally positive, such as fraud detection and e-commerce. When features are not strictly positive, a simple shift or min-max scaling transformation can be applied to ensure positivity. The result establishes signomial classifiers as a theoretically sound alternative to neural networks, which also possess universal approximation properties (Cybenko, 1989; Hornik et al., 1989). While neural networks achieve universal approximation through compositions of sigmoidal activation functions, our result is specifically tailored to the positive orthant where multiplicative power-law relationships are prevalent. The theorem guarantees that signomial classifiers can approximate any continuous decision boundary to arbitrary precision given sufficient terms.

*Proof.* We transform to log-space to reduce the problem to the classical Stone-Weierstrass theorem.

Define the transformation $\phi : \mathbb{R}^n_{>0} \to \mathbb{R}^n$ by

$$\phi(x_1, \ldots, x_n) = (y_1, \ldots, y_n) \qquad \text{where} \quad y_j = \log x_j \quad \text{for } j = 1, \ldots, n. \tag{A1}$$

This is a homeomorphism with inverse $\phi^{-1}(y_1, \ldots, y_n) = (e^{y_1}, \ldots, e^{y_n})$. Let $\tilde{D} = \phi(D)$. Since $D$ is compact and $\phi$ is continuous, $\tilde{D}$ is compact in $\mathbb{R}^n$. Define $g : \tilde{D} \to \mathbb{R}$ by

$$g(y_1, \ldots, y_n) = f(e^{y_1}, \ldots, e^{y_n}).$$

Since $f$ is continuous, $g$ is continuous on $\tilde{D}$.

Consider a signomial $S(x_1, \ldots, x_n) = \sum_{k=1}^{K} \alpha_k \prod_{j=1}^{n} x_j^{\beta_{kj}}$. Substituting $x_j = e^{y_j}$, we obtain

$$S(e^{y_1}, \ldots, e^{y_n}) = \sum_{k=1}^{K} \alpha_k \prod_{j=1}^{n} e^{\beta_{kj} y_j} = \sum_{k=1}^{K} \alpha_k e^{\sum_{j=1}^{n} \beta_{kj} y_j}. \tag{A2}$$

Writing $\langle \beta_k, y \rangle = \sum_{j=1}^{n} \beta_{kj} y_j$ for the inner product in Eq. (A2), signomials in the original space correspond to finite linear combinations of functions of the form $e^{\langle \beta, y \rangle}$ in log-space. Let $\mathcal{A}$ be the set of all such finite linear combinations:

$$\mathcal{A} = \left\{ h : h(y_1, \ldots, y_n) = \sum_{k=1}^{K} \alpha_k e^{\langle \beta_k, y \rangle}, \quad K \in \mathbb{N}, \ \alpha_k \in \mathbb{R}, \ \beta_k \in \mathbb{R}^n \right\}.$$

We verify that $\mathcal{A}$ satisfies the conditions of the Stone-Weierstrass theorem. First, $\mathcal{A}$ is an algebra: it is clearly closed under addition and scalar multiplication. For multiplication, observe that

$$e^{\langle \beta, y \rangle} \cdot e^{\langle \delta, y \rangle} = e^{\langle \beta + \delta, y \rangle},$$

so products of functions in $\mathcal{A}$ remain in $\mathcal{A}$ since $\beta + \delta \in \mathbb{R}^n$. Second, $\mathcal{A}$ contains constants: taking $\beta = 0$ gives $e^{\langle 0, y \rangle} = 1$. Third, $\mathcal{A}$ separates points: if $(y_1, \ldots, y_n) \neq (y_1', \ldots, y_n')$, then $y_j \neq y_j'$ for some $j$. Taking $\beta = e_j$ (the $j$-th standard basis vector), we have $e^{y_j} \neq e^{y_j'}$ since the exponential is injective.

By the Stone-Weierstrass theorem, $\mathcal{A}$ is dense in $C(\tilde{D}, \mathbb{R})$. And so, for any $\epsilon > 0$, there exists $h(y_1, \ldots, y_n) = \sum_{k=1}^{K} \alpha_k e^{\langle \beta_k, y \rangle} \in \mathcal{A}$ such that

$$\sup_{(y_1, \ldots, y_n) \in \tilde{D}} |g(y_1, \ldots, y_n) - h(y_1, \ldots, y_n)| < \epsilon.$$

Transforming back to the original coordinates via the substitution in Eq. (A1), we obtain

$$h(\log x_1, \ldots, \log x_n) = \sum_{k=1}^{K} \alpha_k \prod_{j=1}^{n} e^{\beta_{kj} \log x_j} = \sum_{k=1}^{K} \alpha_k \prod_{j=1}^{n} x_j^{\beta_{kj}} =: S(x_1, \ldots, x_n),$$

which is a signomial. For any $(x_1, \ldots, x_n) \in D$ with $(y_1, \ldots, y_n) = \phi(x_1, \ldots, x_n)$, we have

$$|f(x_1, \ldots, x_n) - S(x_1, \ldots, x_n)| = |g(y_1, \ldots, y_n) - h(y_1, \ldots, y_n)| < \epsilon.$$

Therefore $\sup_{(x_1, \ldots, x_n) \in D} |f(x_1, \ldots, x_n) - S(x_1, \ldots, x_n)| < \epsilon.$ $\qquad\square$

## B. Faithfulness of Exponent Magnitude

This section establishes that for signomial functions with $K = 1$, exponent magnitude faithfully reflects feature contribution under proportional perturbations.

**Proposition B.1.** *Let $z(x) = \alpha \prod_{j=1}^{m} x_j^{\beta_j}$ be a signomial with $x_j > 0, \beta_j \in \mathbb{R}$, and $\alpha \neq 0$. Let $r > 0$ be a fixed scaling factor and $r \neq 1$. For any two features $x_k$ and $x_\ell$, define the marginal contribution $\Delta_k(r)$ of a feature $x_k$ as the magnitude of the output response induced by scaling that feature by $r$, holding all other features fixed. Then it holds that*

$$|\beta_k| > |\beta_\ell| \implies \Delta_k(r) > \Delta_\ell(r).$$

*In other words, the magnitude of the exponent of a signomial faithfully reflects the magnitude of a feature's contribution to the output.*

*Proof.* Suppose we scale feature $x_k$ by factor $r$ and keep all other features fixed, then

$$z_{\text{new}} = \alpha (r \cdot x_k)^{\beta_k} \prod_{j \neq k} x_j^{\beta_j} = r^{\beta_k} \cdot z \tag{A3}$$

Thus scaling $x_k$ by $r$ scales the output by factor $r^{\beta_k}$. Since $\alpha$ may be negative, we measure the output magnitude using absolute values. From (A3),

$$|z_{\text{new}}| = |r^{\beta_k} \cdot z| = r^{\beta_k} \cdot |z|.$$

Since $\alpha \neq 0$ and $x_j > 0$ for all $j$, we have $z \neq 0$ and thus $|z| > 0$ (and likewise $|z_{\text{new}}| > 0$), so $\log |z|$ and $\log |z_{\text{new}}|$ are well-defined. Taking logarithms converts the multiplicative change in the output magnitude into an additive quantity, allowing us to measure the strength of the proportional response. Thus,

$$\log |z_{\text{new}}| = \log(r^{\beta_k} \cdot |z|) \iff \log |z_{\text{new}}| - \log |z| = \beta_k \log r \tag{A4}$$

And so we can define the marginal contribution of feature $x_k$ under scaling factor $r$ as:

$$\Delta_k(r) := |\log |z_{\text{new}}| - \log |z||  \tag{A5}$$

For features $x_k$ and $x_\ell$, it follows from (A4) and (A5) that we have $\Delta_k(r) = |\beta_k| |\log r|$ and $\Delta_\ell(r) = |\beta_\ell| |\log r|$.

Since $r \neq 1$, we have $|\log r| > 0$, and hence

$$|\beta_k| > |\beta_\ell| \implies |\beta_k| |\log r| > |\beta_\ell| |\log r|$$
$$\implies \Delta_k(r) > \Delta_\ell(r).$$

$\qquad\square$

## C. Desirable Properties of ECSEL

This appendix provides complete derivations for Properties G1–G3 (global feature behavior), D1–D2 (decision-level effects), and L1–L2 (local feature attributions) stated in Section 3.2. We prove Theorem 3.2 by showing that ECSEL's signomial structure satisfies each property through closed-form expressions for elasticities, log-gradients, margin sensitivities, probability competition, and feature attributions. We restate the theorem below.

**Theorem** 3.2 (Interpretability properties of ECSEL). *Let $\{(x_i, y_i)\}_{i=1}^N$ be a dataset with $x_i \in \mathbb{R}^m_{>0}$ denoting the preprocessed positive feature representation. Let ECSEL learn class score functions*

$$z_c(x) = \sum_{k=1}^{K} \alpha_{c,k} \prod_{j=1}^{m} x_j^{\beta_{c,k,j}}, \qquad c \in \{0, \ldots, C-1\},$$

*with component scores $z_{c,k}(x) = \alpha_{c,k} \prod_{j=1}^{m} x_j^{\beta_{c,k,j}}$ and log-gradients $G_{c,j}(x) = \sum_{k=1}^{K} \beta_{c,k,j} z_{c,k}(x)$. Then, for all $x \in \mathbb{R}^m_{>0}$ with $z_c(x) > 0$ where required, the resulting classifier satisfies Properties G1–G3, D1–D2, and L1–L2, with the understanding that Properties involving logarithmic derivatives are defined on the natural domain where the corresponding scores are positive.*

*Proof.* We show that each property holds by deriving closed-form expressions that follow directly from ECSEL's signomial structure.

PROPERTY G1: DIRECT GLOBAL FEATURE ATTRIBUTION
We derive the elasticity (proportional sensitivity) of class scores with respect to each feature. The elasticity is defined as

$$E_{c,j}(x) := \frac{\partial \log z_c(x)}{\partial \log x_j}.$$

This quantity is well-defined for inputs such that $z_c(x) > 0$. Using the chain rule to convert from standard derivatives to log-derivatives:

$$E_{c,j}(x) = \frac{\partial \log z_c(x)}{\partial z_c(x)} \cdot \frac{\partial z_c(x)}{\partial x_j} \cdot \frac{\partial x_j}{\partial \log x_j} = \frac{1}{z_c(x)} \cdot \frac{\partial z_c(x)}{\partial x_j} \cdot x_j. \tag{A6}$$

For ECSEL, the class score is $z_c(x) = \sum_{k=1}^{K} z_{c,k}(x)$ where $z_{c,k}(x) = \alpha_{c,k} \prod_{j=1}^{m} x_j^{\beta_{c,k,j}}$. Taking the partial derivative with respect to $x_j$ using the power rule:

$$\frac{\partial z_c(x)}{\partial x_j} = \sum_{k=1}^{K} \frac{\partial z_{c,k}(x)}{\partial x_j} = \sum_{k=1}^{K} \alpha_{c,k} \beta_{c,k,j} x_j^{\beta_{c,k,j}-1} \prod_{\ell \neq j} x_\ell^{\beta_{c,k,\ell}} = \sum_{k=1}^{K} \frac{\beta_{c,k,j}}{x_j} z_{c,k}(x).$$

Substituting into Eq. (A6):

$$E_{c,j}(x) = \frac{1}{z_c(x)} \cdot \sum_{k=1}^{K} \frac{\beta_{c,k,j}}{x_j} z_{c,k}(x) \cdot x_j = \frac{1}{z_c(x)} \sum_{k=1}^{K} \beta_{c,k,j} z_{c,k}(x) = \sum_{k=1}^{K} \frac{z_{c,k}(x)}{z_c(x)} \beta_{c,k,j},$$

which is the equation defined in Eq. (6). For $K = 1$, this simplifies to $E_{c,j}(x) = \beta_{c,j}$, a constant elasticity, since the weight $\frac{z_{c,1}(x)}{z_c(x)} = 1$ when there is only a single component. For $K > 1$, the elasticity is a weighted combination of component-specific exponents, with weights given by the relative component contributions $\frac{z_{c,k}(x)}{z_c(x)}$.

PROPERTY G2: EXACT COUNTERFACTUAL REASONING
We derive the effect of scaling feature $x_j$ by a factor $q > 0$ on the class score. Let $x^{\text{new}}$ denote the perturbed input where $x_j^{\text{new}} = q \cdot x_j$ and $x_\ell^{\text{new}} = x_\ell$ for all $\ell \neq j$.

For each component $k$, the new component score is:

$$
\begin{aligned}
z_{c,k}(x^{\text{new}}) &= \alpha_{c,k} \prod_{\ell=1}^{m} (x_\ell^{\text{new}})^{\beta_{c,k,\ell}} \\
&= \alpha_{c,k}(q \cdot x_j)^{\beta_{c,k,j}} \prod_{\ell \neq j} x_\ell^{\beta_{c,k,\ell}} \\
&= \alpha_{c,k} q^{\beta_{c,k,j}} x_j^{\beta_{c,k,j}} \prod_{\ell \neq j} x_\ell^{\beta_{c,k,\ell}} = q^{\beta_{c,k,j}} z_{c,k}(x),
\end{aligned}
\tag{A7}
$$

where the last equality follows from recognizing that $\alpha_{c,k} x_j^{\beta_{c,k,j}} \prod_{\ell \neq j} x_\ell^{\beta_{c,k,\ell}} = z_{c,k}(x)$.

Summing Eq. (A7) over all $K$ components to obtain the total class score, gives:

$$
z_c(x^{\text{new}}) = \sum_{k=1}^{K} z_{c,k}(x^{\text{new}}) = \sum_{k=1}^{K} q^{\beta_{c,k,j}} z_{c,k}(x).
$$

This closed-form expression, as first shown in Eq. (7), allows exact computation of counterfactual scores without re-evaluation of the model. For $K = 1$, the sum contains only a single term, so:

$$
z_c(x^{\text{new}}) = q^{\beta_{c,1,j}} z_{c,1}(x) = q^{\beta_{c,j}} z_c(x),
$$

where we simplify notation by dropping the component index when $K = 1$. This shows that the score scales by a power of the perturbation factor determined solely by the exponent $\beta_{c,j}$.

PROPERTY G3: GRADIENT-BASED SENSITIVITY

We derive the first-order response of class scores to small proportional changes in features. Consider scaling feature $x_j$ by a small factor $(1 + \varepsilon)$ where $|\varepsilon| \ll 1$. From Property G2, we have:

$$
z_c(x^{\text{new}}) = \sum_{k=1}^{K} (1 + \varepsilon)^{\beta_{c,k,j}} z_{c,k}(x).
$$

Applying the first-order Taylor expansion $(1 + \varepsilon)^\beta \approx 1 + \beta\varepsilon + \mathcal{O}(\varepsilon^2)$ as $\varepsilon \to 0$:

$$
\begin{aligned}
z_c(x^{\text{new}}) &= \sum_{k=1}^{K} (1 + \beta_{c,k,j}\varepsilon + \mathcal{O}(\varepsilon^2)) z_{c,k}(x) \\
&= \sum_{k=1}^{K} z_{c,k}(x) + \varepsilon \sum_{k=1}^{K} \beta_{c,k,j} z_{c,k}(x) + \mathcal{O}(\varepsilon^2) \\
&= z_c(x) + \varepsilon \sum_{k=1}^{K} \beta_{c,k,j} z_{c,k}(x) + \mathcal{O}(\varepsilon^2).
\end{aligned}
$$

Recognizing that $\sum_{k=1}^{K} \beta_{c,k,j} z_{c,k}(x) = G_{c,j}(x)$, we obtain:

$$
z_c(x^{\text{new}}) = z_c(x) + \varepsilon \cdot G_{c,j}(x) + \mathcal{O}(\varepsilon^2).
$$

By definition, $G_{c,j}(x) = \frac{\partial z_c(x)}{\partial \log x_j}$, confirming that the log-gradient provides exact first-order sensitivity. Furthermore, using Property G1, we have $G_{c,j}(x) = z_c(x) \cdot E_{c,j}(x)$, giving:

$$
z_c(x^{\text{new}}) = z_c(x) + \varepsilon \cdot z_c(x) \cdot E_{c,j}(x) + \mathcal{O}(\varepsilon^2) = z_c(x)(1 + \varepsilon \cdot E_{c,j}(x)) + \mathcal{O}(\varepsilon^2).
$$

For $K = 1$, the elasticity is constant ($E_{c,j}(x) = \beta_{c,j}$ from Property G1), so the response simplifies to

$$
z_c(x^{\text{new}}) = z_c(x)(1 + \varepsilon\beta_{c,j}) + \mathcal{O}(\varepsilon^2),
$$

showing that the score scales proportionally with the exponent.

PROPERTY D1: DECISION BOUNDARY SENSITIVITY

We derive the sensitivity of the decision boundary between classes $c$ and $c'$ to proportional feature changes. The decision boundary is characterized by the margin $z_c(x) - z_{c'}(x)$, and its sensitivity to feature $x_j$ is given by:

$$\frac{\partial}{\partial \log x_j}\big(z_c(x) - z_{c'}(x)\big).$$

Using the linearity of differentiation:

$$\frac{\partial}{\partial \log x_j}\big(z_c(x) - z_{c'}(x)\big) = \frac{\partial z_c(x)}{\partial \log x_j} - \frac{\partial z_{c'}(x)}{\partial \log x_j} = G_{c,j}(x) - G_{c',j}(x).$$

Applying Property G1, we know that $G_{c,j}(x) = z_c(x) \cdot E_{c,j}(x)$ and $G_{c',j}(x) = z_{c'}(x) \cdot E_{c',j}(x)$. And so we obtain Eq. (10):

$$\frac{\partial}{\partial \log x_j}\big(z_c(x) - z_{c'}(x)\big) = z_c(x) \cdot E_{c,j}(x) - z_{c'}(x) \cdot E_{c',j}(x).$$

For $K = 1$, the elasticities are constant ($E_{c,j}(x) = \beta_{c,j}$ and $E_{c',j}(x) = \beta_{c',j}$ from Property G1), so this simplifies to:

$$\frac{\partial}{\partial \log x_j}\big(z_c(x) - z_{c'}(x)\big) = z_c(x)\beta_{c,j} - z_{c'}(x)\beta_{c',j}.$$

This closed-form expression directly reveals how exponent differences between classes drive the competitive effect of each feature on the decision boundary. A feature pushes the boundary toward class $c$ when $z_c(x)\beta_{c,j} > z_{c'}(x)\beta_{c',j}$.

PROPERTY D2: DIRECT PROBABILITY COMPETITION

We derive the sensitivity of predicted class probabilities to proportional feature changes. For a softmax probability $p_c(x) = \frac{e^{z_c(x)}}{\sum_{r=0}^{C-1} e^{z_r(x)}}$, we compute:

$$\frac{\partial p_c(x)}{\partial \log x_j}.$$

Using the chain rule and the quotient rule for the softmax:

$$\begin{aligned}
\frac{\partial p_c(x)}{\partial \log x_j} &= \frac{\partial}{\partial \log x_j}\left(\frac{e^{z_c(x)}}{\sum_{r=0}^{C-1} e^{z_r(x)}}\right) \\
&= \frac{e^{z_c(x)}\frac{\partial z_c(x)}{\partial \log x_j}\sum_r e^{z_r(x)} - e^{z_c(x)}\sum_r e^{z_r(x)}\frac{\partial z_r(x)}{\partial \log x_j}}{(\sum_r e^{z_r(x)})^2} \\
&= \frac{e^{z_c(x)}}{\sum_r e^{z_r(x)}}\left(\frac{\partial z_c(x)}{\partial \log x_j} - \frac{\sum_r e^{z_r(x)}\frac{\partial z_r(x)}{\partial \log x_j}}{\sum_r e^{z_r(x)}}\right) \\
&= p_c(x)\left(\frac{\partial z_c(x)}{\partial \log x_j} - \sum_r p_r(x)\frac{\partial z_r(x)}{\partial \log x_j}\right).
\end{aligned} \tag{A8}$$

Substituting the log-gradients $G_{c,j}(x) = \frac{\partial z_c(x)}{\partial \log x_j}$ and $G_{r,j}(x) = \frac{\partial z_r(x)}{\partial \log x_j}$ into Eq. (A8) gives Eq. (11):

$$\frac{\partial p_c(x)}{\partial \log x_j} = p_c(x)\big(G_{c,j}(x) - \sum_r p_r(x)G_{r,j}(x)\big).$$

This shows that a feature increases class $c$ probability only when its log-gradient for class $c$ exceeds the probability-weighted average log-gradient across all classes. The competitive nature arises from the normalization constraint $\sum_c p_c(x) = 1$: increasing one class probability necessarily decreases others.

For $K = 1$, using Property G1, the log-gradients are $G_{c,j}(x) = z_c(x)\beta_{c,j}$, so:

$$\frac{\partial p_c(x)}{\partial \log x_j} = p_c(x)\Big(z_c(x)\beta_{c,j} - \sum_r p_r(x)z_r(x)\beta_{r,j}\Big).$$

PROPERTY L1: EXACT LOG-SPACE ADDITIVITY

We derive the exact additive decomposition of individual signomial components in log-space. For each component $k$ of class $c$, the component score is:

$$z_{c,k}(x) = \alpha_{c,k} \prod_{j=1}^m x_j^{\beta_{c,k,j}}.$$

Taking logarithms:

$$\log z_{c,k}(x) = \log \left( \alpha_{c,k} \prod_{j=1}^m x_j^{\beta_{c,k,j}} \right)$$

$$= \log \alpha_{c,k} + \sum_{j=1}^m \beta_{c,k,j} \log x_j. \tag{A9}$$

Let $b \in \mathbb{R}_{>0}^m$ denote a reference baseline (e.g., the geometric mean of the dataset or a specific reference instance). We can rewrite Eq. (A9) as:

$$\log z_{c,k}(x) = \log \alpha_{c,k} + \sum_{j=1}^m \beta_{c,k,j} \log b_j + \sum_{j=1}^m \beta_{c,k,j}(\log x_j - \log b_j)$$

$$= \log z_{c,k}(b) + \sum_{j=1}^m \beta_{c,k,j} \log \frac{x_j}{b_j},$$

where $z_{c,k}(b) = \alpha_{c,k} \prod_{j=1}^m b_j^{\beta_{c,k,j}}$ is the component score at the baseline.

This exact additive decomposition shows that each feature $j$ contributes $\beta_{c,k,j} \log \frac{x_j}{b_j}$ to the log-score of component $k$, yielding an exact additive attribution in log-space that is structurally analogous to Shapley-value explanations (Lundberg & Lee, 2017).

For $K = 1$, the same log-space decomposition holds for the class log-score $\log z_c(x)$, corresponding to an exact additive decomposition of the log-score:

$$\log z_c(x) = \log z_c(b) + \sum_{j=1}^m \beta_{c,j} \log \frac{x_j}{b_j}.$$

For $K > 1$, the aggregated score $z_c(x) = \sum_{k=1}^K z_{c,k}(x)$ is a nonlinear sum of exponentials in log-space, so exact additive decomposition is not available at the aggregate level. However, the log-space decomposition remains exact at the component level, and for the aggregated score, we use gradient-based local approximations (Property L2).

PROPERTY L2: GRADIENT-BASED LOCAL ATTRIBUTIONS

We derive gradient-based local feature attributions for aggregated nonlinear functions where exact additive decomposition is unavailable. For ECSEL with $K > 1$, the class score $z_c(x) = \sum_{k=1}^K z_{c,k}(x)$ is a sum of exponentials in log-space, which is nonlinear and does not admit exact additive decomposition.

We use a first-order Taylor approximation around a baseline $x^* \in \mathbb{R}_{>0}^m$ to obtain local gradient-based feature attributions.

Expanding $z_c(x)$ in log-space:

$$\log z_c(x) \approx \log z_c(x^*) + \sum_{j=1}^{m} \frac{\partial \log z_c(x^*)}{\partial \log x_j}(\log x_j - \log x_j^*)$$

$$= \log z_c(x^*) + \sum_{j=1}^{m} E_{c,j}(x^*)(\log x_j - \log x_j^*),$$

where we use the elasticity $E_{c,j}(x^*) = \frac{\partial \log z_c(x^*)}{\partial \log x_j}$ from Property G1.

From G1, we have $E_{c,j}(x^*) = \frac{G_{c,j}(x^*)}{z_c(x^*)}$, so the contribution of feature $j$ to the log-score is:

$$\phi_j^{(\log z_c)}(x) \approx E_{c,j}(x^*)(\log x_j - \log x_j^*) = \frac{G_{c,j}(x^*)}{z_c(x^*)}(\log x_j - \log x_j^*).$$

Alternatively, we can work directly with the score $z_c(x)$ rather than its logarithm:

$$z_c(x) \approx z_c(x^*) + \sum_{j=1}^{m} \frac{\partial z_c(x^*)}{\partial \log x_j}(\log x_j - \log x_j^*)$$

$$= z_c(x^*) + \sum_{j=1}^{m} G_{c,j}(x^*)(\log x_j - \log x_j^*),$$

giving the contribution:

$$\phi_j^{(z_c)}(x) \approx G_{c,j}(x^*)(\log x_j - \log x_j^*).$$

An analogous procedure applies to class probabilities. Using the probability gradient from Property D2:

$$p_c(x) \approx p_c(x^*) + \sum_{j=1}^{m} \frac{\partial p_c(x^*)}{\partial \log x_j}(\log x_j - \log x_j^*),$$

where $\frac{\partial p_c(x^*)}{\partial \log x_j} = p_c(x^*)\big(G_{c,j}(x^*) - \sum_r p_r(x^*)G_{r,j}(x^*)\big)$ from Property D2. This gives the feature contribution to probability:

$$\phi_j^{(p_c)}(x) \approx p_c(x^*)\big(G_{c,j}(x^*) - \sum_r p_r(x^*)G_{r,j}(x^*)\big)(\log x_j - \log x_j^*).$$

These gradient-based attributions provide local explanations through closed-form expressions, avoiding the computational expense of sampling-based methods such as KernelSHAP (Lundberg & Lee, 2017). The approximation quality depends on the locality of the baseline: for small deviations $\|x - x^*\|$, the first-order Taylor expansion is accurate. $\qquad\square$

# D. Symbolic Regression Benchmark

This appendix provides the technical specifications and results for the symbolic regression experiments presented in Section 4. In the following sections, we detail the data generation procedures, the specific hyperparameter configurations for all baseline models, and provide all results, including a qualitative analysis of cases where ECSEL achieves high numerical fidelity despite structural mismatch.

## D.1. Data Generation and Sampling

To ensure a fair comparison, all methods are evaluated on identical datasets for each target equation. Data generation follows the standard protocols established in the symbolic regression literature:

- **Sampling:** For each equation, we uniformly sample between 20 and 1,000 points.

- **Input Ranges:** Features are drawn from task-specific ranges consistent with the original DGSR benchmark, including $[-5, 5]$, $[-50, 50]$, $[1, 5]$, and $[-10, 10]$.

- **Noise:** We add Gaussian noise with $\sigma = 0.01$ to the target values to assess the robustness of recovery under realistic conditions.

## D.2. Baseline Implementation Details

All baseline methods are executed using the DGSR authors' publicly available benchmarking suite to ensure consistent data handling. We impose a strict **15-minute time limit** per equation for all methods. Specific baseline configurations are as follows:

- **NeSymRes:** Consistent with its pre-training distribution, this model is evaluated only on equations containing at most three variables.

- **DGSR:** We use the authors' strongest configuration with a beam size of 256. For equations involving the constant $\pi$, the maximum token length is increased from 30 to 50 to ensure the expressions are representable within the search space.

- **NGGP:** Run with default parameters as specified in the benchmark suite.

In Table 3, entries marked "–" indicate cases where a method is not applicable (e.g., NeSymRes on problems with more than three variables), while "dnf" denotes runs that did not finish within the time limit.

## D.3. Full Benchmark Results

The comprehensive performance evaluation across all 58 target expressions is detailed in Table 3. These equations span a wide range of structural complexities, from simple univariate monomials to multi-term signomials with rational exponents. Aggregated performance metrics, including mean recovery rates and computational efficiency, are summarized in the final row of the table.

*Table 3.* Comparative performance on symbolic recovery tasks. Columns report the success rate for exact symbolic identification and the mean solving time across multiple random seeds.

| Eq. | Expression | ECSEL | | DGSR | | NGGP | | NeSymRes | |
|---|---|---|---|---|---|---|---|---|---|
| | | Rec. (%) | Time (s) | Rec. (%) | Time (s) | Rec. (%) | Time (s) | Rec. (%) | Time (s) |
| I.12.1 | $F = \mu N n$ | 100 | $0.75 \pm 0.67$ | 100 | $7.72 \pm 2.87$ | 100 | $1.27 \pm 0.16$ | 100 | $35.42 \pm 1.29$ |
| I.12.2 | $F = \frac{q_1 q_2}{4\pi\epsilon r^2}$ | 100 | $0.15 \pm 0.01$ | 27.8 | $739.39 \pm 309.09$ | 35.3 | $137.20 \pm 67.64$ | – | – |
| I.12.4 | $E_f = \frac{q_1}{4\pi\epsilon r^2}$ | 100 | $0.10 \pm 0.02$ | 0 | $600.61 \pm 4.49$ | 0 | $148.56 \pm 0.90$ | – | – |
| I.12.5 | $F = q_2 E_f$ | 100 | $0.04 \pm 0.01$ | 100 | $5.79 \pm 1.01$ | 80 | $46.17 \pm 90.01$ | 100 | $34.37 \pm 2.06$ |
| I.14.3 | $U = mgz$ | 100 | $0.75 \pm 0.70$ | 100 | $4.12 \pm 0.53$ | 100 | $0.86 \pm 0.08$ | 100 | $27.66 \pm 2.40$ |
| I.14.4 | $U = k_{spring} x^2$ | 100 | $0.04 \pm 0.01$ | 100 | $2.84 \pm 0.09$ | 100 | $0.65 \pm 0.06$ | 100 | $23.36 \pm 0.52$ |

| Eq. | Expression | ECSEL | | DGSR | | NGGP | | NeSymRes | |
|---|---|---|---|---|---|---|---|---|---|
| | | Rec. (%) | Time (s) | Rec. (%) | Time (s) | Rec. (%) | Time (s) | Rec. (%) | Time (s) |
| I.25.13 | $V_e = \dfrac{q}{C}$ | 100 | 0.06 ± 0.06 | 100 | 3.52 ± 1.10 | 100 | 0.81 ± 0.10 | 100 | 22.51 ± 0.26 |
| I.29.4 | $k = \dfrac{\omega}{c}$ | 100 | 0.06 ± 0.06 | 100 | 2.98 ± 0.15 | 100 | 0.84 ± 0.23 | 100 | 22.39 ± 0.13 |
| I.32.5 | $P = \dfrac{q^2 a^2}{6\pi\epsilon c^3}$ | 100 | 0.28± 0.02 | 0 | 591.77 ± 6.02 | 0 | 144.61 ± 0.83 | – | – |
| I.34.8 | $\omega = \dfrac{qvB}{p}$ | 100 | 0.57 ± 0.50 | 100 | 9.99 ± 0.00 | 100 | 1.49 ± 0.00 | – | – |
| I.34.27 | $E = \hbar\omega$ | 100 | 0.06 ± 0.06 | 100 | 3.48 ± 1.00 | 100 | 0.82 ± 0.07 | 100 | 22.78 ± 0.27 |
| I.38.12 | $r = \dfrac{4\pi\epsilon\hbar^2}{mq^2}$ | 100 | 0.10 ± 0.05 | 0 | 599.62 ± 7.74 | 0 | 151.38 ± 0.89 | – | – |
| I.39.10 | $E = \dfrac{3}{2}pFV$ | 100 | 0.81 ± 0.76 | 100 | 44.56 ±54.64 | 100 | 9.35 ± 5.54 | 0 | 250.22 ± 2.05 |
| I.39.22 | $P_F = \dfrac{nk_bT}{V}$ | 100 | 0.61 ± 0.54 | 100 | 15.85 ± 0 | 100 | 1.55 ± 0 | – | – |
| I.43.16 | $v = \dfrac{\mu_{drift}qV_e}{d}$ | 100 | 0.60 ± 0.52 | 100 | 10.79 ± 0.28 | 100 | 1.99 ± 0.28 | – | – |
| I.43.31 | $D = \mu_e k_b T$ | 100 | 0.85 ± 0.79 | 100 | 4.12 ± 0.96 | 100 | 0.69 ± 0.05 | 100 | 25.99 ± 1.29 |
| I.47.23 | $c = \sqrt{\dfrac{\gamma pr}{\rho}}$ | 100 | 0.28 ± 0.34 | 0 | 681.89 ±54.90 | 0 | 148.84 ± 4.49 | 0 | 260.26 ±14.51 |
| II.3.24 | $F_E = \dfrac{P}{4\pi r^2}$ | 100 | 0.15 ± 0.24 | 0 | dnf | 0 | 147.17 ± 3.90 | 0 | 255.24 ± 1.02 |
| II.4.23 | $V_e = \dfrac{q}{4\pi\epsilon r}$ | 100 | 0.03 ± 0.02 | 0 | 574.11 ± 9.35 | 0 | 161.83 ± 5.51 | 0 | 262.06 ± 4.22 |
| II.8.7 | $E = \dfrac{3}{5}\dfrac{q^2}{4\pi\epsilon r}$ | 100 | 0.38 ±0.67 | 100 | 618.94 ±32.38 | 0 | 149.13 ± 2.59 | 0 | 253.97 ± 5.09 |
| II.8.31 | $E_{den} = \epsilon E_f^2$ | 100 | 0.04 ± 0.01 | 100 | 2.83 ± 0.13 | 100 | 0.60 ± 0.04 | 100 | 23.10 ± 0.14 |
| II.11.20 | $P_* = \dfrac{n_p p_d^2 E_f}{3k_bT}$ | 100 | 0.04 ± 0.02 | 100 | 393.90 ± 0.00 | 100 | 19.61 ± 0.00 | – | – |
| II.13.17 | $B = \dfrac{1}{4\pi\epsilon c^2}\dfrac{2I}{r}$ | 100 | 0.14 ± 0.05 | 0 | 594.28 ± 3.08 | 0 | 148.13 ± 1.63 | – | – |
| II.27.16 | $F_E = \epsilon c E_f^2$ | 100 | 0.69 ± 0.59 | 100 | 3.47 ± 0.94 | 100 | 0.81 ± 0.08 | 100 | 23.24 ± 0.81 |
| II.27.18 | $E_{den} = \epsilon E_f^2$ | 100 | 0.04 ± 0.01 | 100 | 2.85 ± 0.06 | 100 | 23.44 ± 0.79 | 100 | 0.60 ± 0.02 |
| II.34.2A | $I = \dfrac{qv}{2\pi r}$ | 100 | 0.70 ± 0.78 | 0 | 661.94 ±26.74 | 0 | 155.48 ± 3.36 | 0 | 274.73 ± 6.11 |
| II.34.2 | $\mu_M = \dfrac{qvr}{2m}$ | 100 | 0.55 ± 0.48 | 100 | 145.80 ±32.03 | 100 | 25.51 ±33.91 | – | – |
| II.34.11 | $\omega = \dfrac{gqB}{2m}$ | 100 | 0.52 ± 0.45 | 100 | 121.01 ±97.13 | 100 | 4.04 ±16.88 | – | – |
| II.34.29A | $\mu_M = \dfrac{qh}{4\pi m}$ | 100 | 0.39 ± 0.41 | 0 | 768.60 ± 0.00 | 0 | 161.46 ± 0.00 | 0 | 263.47 ± 8.96 |
| II.34.29B | $E = g\mu_M B J_z$ | 100 | 0.12 ± 0.15 | 100 | 9.12 ± 6.79 | 100 | 1.57 ± 1.27 | – | – |
| II.38.3 | $E = \dfrac{YAx}{d}$ | 100 | 0.58 ± 0.51 | 100 | 10.40 ± 3.72 | 100 | 0.94 ± 0.26 | – | – |
| III.7.38 | $\omega = \dfrac{2\mu_M B}{\hbar}$ | 100 | 0.70 ± 0.60 | 100 | 3.45 ± 1.11 | 100 | 0.91 ± 0.28 | 100 | 22.10 ± 0.17 |
| III.12.43 | $L = n\hbar$ | 100 | 0.04 ± 0.01 | 100 | 3.30 ± 0.81 | 100 | 0.86 ± 0.08 | 100 | 23.72 ± 0.56 |
| III.13.18 | $v = \dfrac{2Ed^2 k}{\hbar}$ | 100 | 0.08 ± 0.05 | 100 | 8.27 ± 3.26 | 100 | 5.19 ± 1.84 | – | – |
| III.15.14 | $m = \dfrac{\hbar^2}{2Ed^2}$ | 100 | 0.78 ± 0.65 | 100 | 59.10 ±34.71 | 100 | 3.76 ± 2.57 | – | – |
| III.15.27 | $k = \dfrac{2\pi\alpha}{nd}$ | 100 | 0.64 ± 0.54 | 0 | 598.85 ± 4.46 | 0 | 152.33 ± 1.20 | – | – |
| III.19.51 | $E = -\dfrac{mq^4}{2(4\pi\epsilon)^2\hbar^2}\dfrac{1}{n^2}$ | 100 | 0.46 ± 0.10 | 0 | 596.54 ± 4.33 | 0 | 150.20 ± 0.81 | – | – |
| III.21.20 | $j = -\dfrac{\rho_c q A v_e c}{m}$ | 100 | 0.03 ± 0.02 | 100 | 23.38 ±13.50 | 100 | 14.59 ± 5.17 | – | – |
| Livermore-12 | $x_1^5 \cdot x_2^{-3}$ | 100 | 0.07 ± 0.02 | 100 | 12.35 ± 6.52 | 100 | 6.48 ± 5.50 | 100 | 22.15 ± 0.15 |
| Livermore-13 | $\sqrt[3]{x_1}$ | 100 | 0.05 ± 0.00 | 100 | 41.15 ± 29.89 | 100 | 13.30 ±16.08 | 0 | 246.34 ± 2.10 |
| Livermore-15 | $\sqrt[5]{x_1}$ | 100 | 0.05 ± 0.00 | 100 | 122.03 ±61.11 | 100 | 18.33 ±11.00 | 0 | 251.97 ± 1.10 |
| Livermore-16 | $\sqrt[3]{x_1^2}$ | 100 | 0.05 ± 0.00 | 0 | 749.88 ±211.06 | 80 | 58.82 ±44.48 | 0 | 256.01 ± 9.54 |
| Constant-5 | $\sqrt{1.23x_1}$ | 100 | 0.02 ± 0.00 | 0 | 549.52 ± 0.00 | 0 | 127.55 ± 0.00 | 0 | 253.02 ± 0.00 |
| Constant-6 | $x_1^{0.426}$ | 100 | 0.02 ± 0.00 | 0 | 276.92 ± 58.08 | 0 | 132.13 ± 6.49 | – | – |
| I.11.19 | $A = x_1 y_1 + x_2 y_2 + x_3 y_3$ | 100 | 151.49 ± 16.26 | 100 | 523.01 ± 466.76 | 100 | 17.12 ± 17.77 | – | – |
| I.13.4 | $K = \frac{1}{2}m(v^2 + u^2 + w^2)$ | 100 | 101.33 ± 6.45 | 0 | dnf | 0 | 143.87 ± 1.62 | – | – |
| I.13.12 | $G = m_1 m_2 \left(\frac{1}{r_2} - \frac{1}{r_1}\right)$ | 100 | 58.32 ± 12.54 | 100 | 215.76 ± 133.74 | 100 | 10.98 ± 10.66 | – | – |
| I.24.6 | $E = \frac{1}{4}m(\omega^2 + \omega_0^2)x^2$ | 100 | 68.43 ± 13.93 | 0 | dnf | 0 | 211.60 ± 37.36 | – | – |
| II.2.42 | $P = \dfrac{\kappa(T_2 - T_1)A}{d}$ | 100 | 101.10 ± 8.06 | 100 | 54.15 ± 38.98 | 100 | 1.62 ± 0.88 | – | – |
| II.36.38 | $f = \dfrac{\mu_m B}{k_b T} + \dfrac{\mu_m \alpha M}{\epsilon c^2 k_b T}$ | 40 | 89.11 ± 15.44 | 100 | dnf | 100 | 107.38 ± 31.26 | – | – |
| II.37.1 | $E = \mu_M(1 + \chi)B$ | 100 | 50.67 ± 15.66 | 100 | 21.69 ± 0.17 | 100 | 0.96 ± 0.35 | – | – |
| Jin-1 | $f_1 = 2.5x_1^4 - 1.3x_1^3 + 0.5x_2^2 - 1.7x_2$ | 0 | 53.65 ± 6.83 | 0 | 891.53 ± 32.00 | 0 | 145.61 ± 4.85 | – | – |
| Jin-2 | $f_2 = 8x_1^2 + 8x_2^3 - 15$ | 100 | 92.89 ± 1.96 | 0 | 893.74 ± 44.50 | 0 | 144.18 ± 5.58 | – | – |

| Eq. | Expression | ECSEL | | DGSR | | NGGP | | NeSymRes | |
|-----|------------|-------|---|------|---|------|---|----------|---|
| | | Rec. (%) | Time (s) | Rec. (%) | Time (s) | Rec. (%) | Time (s) | Rec. (%) | Time (s) |
| JIN-3 | $f_3 = 0.2x_1^3 + 0.5x_2^3 - 1.2x_2 - 0.5x_1$ | 100 | $648.46 \pm 14.53$ | 0 | dnf | 0 | $839.02 \pm 196.46$ | – | – |
| KORNS-1 | $y = 1.6 + 24.3x_1$ | 80 | $309.82 \pm 10.80$ | 0 | dnf | 0 | $854.40 \pm 155.09$ | – | – |
| KORNS-2 | $y = 0.23 + 14.2 \cdot \frac{x_4 + x_2}{3 \cdot x_5}$ | 80 | $449.06 \pm 17.78$ | 0 | dnf | 0 | dnf | – | – |
| KORNS-3 | $y = -5.41 + 4.9 \cdot \frac{x_2 - x_1 + \frac{x_3}{x_4}}{x_4}$ | 60 | $497.34 \pm 131.48$ | 0 | dnf | 0 | dnf | – | – |
| KORNS-6 | $y = 1.3 + 0.13\sqrt{x_1}$ | 100 | $74.48 \pm 3.38$ | 0 | dnf | 0 | $830.81 \pm 258.99$ | – | – |
| **Average** | | **95.86** | 86.4 | 59.10 | $612.9^a$ | 58.54 | $468.7^a$ | $56^b$ | $126.3^b$ |

$^a$ Average recovery time assigns 900 seconds to runs marked as "dnf", corresponding to the 15- minute time limit.

$^b$ For NeSymRes, the averages are computed over applicable equations only (25 of 58). Rows marked "–" are excluded, with rows with zero recovery are included.

### D.4. Qualitative Analysis: Numerical Accuracy Without Symbolic Recovery

In a small subset of problems, ECSEL fails to recover the exact symbolic form of the target but achieves near-perfect predictive accuracy ($R^2 \approx 1.0$). This occurs in two distinct scenarios: multi-term expressions with complex structure (found in the benchmark problems of Section 4), and higher-degree univariate polynomials where gradient-based optimization converges to continuous exponent approximations rather than exact integer powers.

**Benchmark Problems.** Table 4 presents equations from the main symbolic regression benchmark (Section 4) where ECSEL's recovery rate falls below 100%. These failures typically occur in multi-term settings where the optimization finds a functional approximation, a structurally distinct signomial that numerically mimics the target over the sampled domain.

*Table 4.* Samples with partial symbolic recovery rate ($< 100\%$). Fitted expression corresponds to one example out of five random seeds that was not recovered. Recovery rate (Rec. %) is the original recovery rate over five seeds.

| Eq. | True Expression | Fitted Expression | Rec. (%) | MSE | NMSE | $R^2$ |
|-----|-----------------|-------------------|----------|-----|------|-------|
| II.36.38 | $f = \frac{\mu_m B}{k_b T} + \frac{\mu_m \alpha M}{\epsilon c^2 k_b T}$ | $0.889 \cdot \frac{x_1^{0.99} x_2^{1.01} x_5^{0.04} x_6^{0.05}}{x_3^{0.01} x_7^{1.00} x_8^{1.00}}$ $+ 1.389 \cdot \frac{x_1^{1.00} x_3^{0.92} x_4^{0.85}}{x_2^{0.02} x_5^{0.99} x_6^{1.92} x_7^{0.99} x_8^{0.98}}$ | 40 | $1.95 \times 10^{-2}$ | 0.00 | 1.00000 |
| JIN-1 | $f_1 = 2.5x_1^4 - 1.3x_1^3 + 0.5x_2^2 - 1.7x_2$ | $f = 2.2 \cdot x_1^4 - 1.1 \cdot x_1^{2.6} + 0.01 \cdot x_2^{2.7} + 2.3 \cdot x_2^{0.9}$ | 0 | $2.3 \times 10^{-5}$ | $1.63 \times 10^{-9}$ | 1.00000 |
| KORNS-1 | $1.6 + 24.3 \cdot x_1$ | $24.300 \cdot x_1 + 1.570$ | 80 | $4.87 \times 10^{-4}$ | $1.04 \times 10^{-6}$ | 1.00000 |
| KORNS-2 | $y = 0.23 + 14.2\frac{x_4 + x_2}{3 \cdot x_5}$ | $3.378 \frac{x_1^{1.12} x_3^{0.06}}{x_2^{0.05} x_4^{1.11}} + 2.927 \cdot \frac{x_1^{0.39} x_2^{0.36}}{x_3^{0.39} x_4^{0.40}}$ $+ 3.403 \cdot \frac{x_2^{1.12} x_4^{0.05}}{x_1^{0.06} x_3^{1.11}}$ | 80 | $5.21 \times 10^{-5}$ | $5.00 \times 10^{-6}$ | 0.999995 |
| KORNS-3 | $y = -5.41 + 4.9\frac{x_2 - x_1 + \frac{x_3}{x_4}}{x_4}$ | $-3.587\frac{x_1^{0.89} x_2^{0.28} x_5^{0.35}}{x_4^{0.21}}$ $+ 2.478\frac{x_2^{0.49} x_3^{0.02} x_4^{0.95} x_5^{0.37}}{x_1^{0.12}}$ $- 2.087\frac{x_1^{0.38}}{x_3^{0.26}} - 1.186\frac{x_1^{0.05}}{x_2^{0.01}}$ | 60 | $3.00 \times 10^{-8}$ | 0.00 | 1.00000 |

**Higher-Degree Univariate Problems.** To assess ECSEL's ability to recover exact integer exponents in higher-degree polynomials, we evaluate on the Nguyen benchmark (Nguyen et al., 2011), a suite of univariate polynomial regression tasks. We generate 100 samples uniformly from $[0, 5]$ with Gaussian noise ($\sigma = 0.01$) added to target values.

ECSEL successfully recovers simple low-degree univariate polynomials: Nguyen-1 ($x + x^2 + x^3$) achieves 100% exact recovery across five random seeds within 3 seconds, and Nguyen-8 ($\sqrt{x}$) reaches 100% recovery within 1 second. However,

as polynomial degree increases, ECSEL struggles to recover the exact symbolic form despite achieving near-perfect numerical fit (MSE $< 10^{-5}$, $R^2 \approx 1.0$). Table 5 shows representative examples where ECSEL learns numerically accurate approximations but fails symbolic recovery. Recovery rates are averaged over five random seeds (42–46), with fitted expressions shown from a single representative seed. The learned expressions exhibit close exponent values (e.g., $x^{2.1}$ vs. $x^2$, $x^{3.1}$ vs. $x^3$) and compensating coefficients that preserve predictive accuracy while deviating from the true symbolic structure.

*Table 5.* ECSEL's learned approximations for higher-degree Nguyen polynomials. Despite near-perfect numerical fit ($R^2 \approx 1.0$, NMSE $\approx 0$), ECSEL fails to recover exact symbolic forms.

| Eq. | True Expression | Fitted Expression | MSE |
|---|---|---|---|
| Nguyen-2 | $x + x^2 + x^3 + x^4$ | $1.4x^{1.1} - 3.4x^{2.8} + 1.1x^{4.0} + 5.0x^{2.7}$ | $3.5 \times 10^{-7}$ |
| Nguyen-3 | $x + x^2 + x^3 + x^4 + x^5$ | $1.0x^{1.0} + 1.0x^{4.1} + 1.0x^{5.0} + 1.1x^{2.1} + 1.0x^{3.1}$ | $1.4 \times 10^{-6}$ |
| Nguyen-4 | $x + x^2 + x^3 + x^4 + x^5 + x^6$ | $1.0x^{1.1} + 1.1x^{1.8} + 1.0x^{4.0} + 1.1x^{6.0} + 0.8x^{2.9} + 1.0x^{4.8}$ | $3.3 \times 10^{-5}$ |

**Non-Signomial Problems.** Beyond polynomial structure, we evaluate ECSEL's approximation quality when the underlying model is non-signomial, we evaluate on eight equations from the AI Feynman and Nguyen benchmarks spanning three non-signomial function classes: trigonometric, rational, and logarithmic. We generate 100 samples per equation with Gaussian noise ($\sigma = 0.01$) and fit signomials with $K \in \{3, 5, 10\}$, reporting the best result per equation.

Table 6 summarizes the results. For smooth, low-oscillation functions, ECSEL achieves near-perfect numerical fidelity despite the structural mismatch. The rational equations I.34.14 and I.39.11 yield $R^2 > 0.998$, and the logarithmic Nguyen-7 achieves $R^2 \approx 1.000$, consistent with the universal approximation result of Theorem 3.1: signomials can approximate any continuous function to arbitrary precision given sufficient terms. For the trigonometric equations, results are more mixed. Equations such as I.18.12, II.15.4, and III.17.37 achieve $R^2 > 0.99$ despite involving sin and cos, suggesting that signomials can approximate weakly oscillatory behavior numerically. For strongly oscillatory functions such as Nguyen-5 ($\sin(x^2)\cos(x) - 1$) and Nguyen-10 ($2\sin(x)\cos(y)$), approximation quality remains poor regardless of $K$, suggesting that gradient-based optimization struggles to find a good approximation within the term budget considered here.

*Table 6.* ECSEL approximation quality on non-signomial equations. For each equation, results from the best-performing $K$ are reported. Despite structural mismatch, ECSEL achieves high numerical fidelity on most equations.

| Eq. | True Expression | Type | $K$ | MSE | $R^2$ |
|---|---|---|---|---|---|
| I.18.12 | $\tau = r \cdot F \cdot \sin(\theta)$ | Trigonometric | 10 | $2.27 \times 10^{-1}$ | 0.996 |
| I.34.14 | $\omega = \omega_0/(1 - v/c)$ | Rational | 3 | $2.22 \times 10^{-2}$ | 0.998 |
| I.39.11 | $E = \frac{1}{\gamma - 1}pV$ | Rational | 10 | $3.77 \times 10^{-3}$ | 0.9995 |
| II.15.4 | $E = -\mu B \cos(\theta)$ | Trigonometric | 10 | $1.60 \times 10^{-1}$ | 0.993 |
| III.17.37 | $f = \beta(1 + \alpha \cos(\theta))$ | Trigonometric | 3 | $1.15 \times 10^{-1}$ | 0.996 |
| Nguyen-5 | $y = \sin(x^2)\cos(x) - 1$ | Trigonometric | 3 | $1.67 \times 10^{-1}$ | 0.094 |
| Nguyen-7 | $y = \log(x + 1) + \log(x^2 + 1)$ | Logarithmic | 10 | $1.38 \times 10^{-4}$ | 0.9999 |
| Nguyen-10 | $y = 2\sin(x)\cos(y)$ | Trigonometric | 10 | $6.26 \times 10^{-2}$ | 0.912 |

# E. Classification Benchmark

This appendix provides the technical specifications, dataset characteristics, and all results for the classification experiments.

## E.1. Datasets and Preprocessing

We evaluate all methods on 11 binary and multi-class datasets. Table 7 provides a description of each dataset and the corresponding classification task.

Categorical features are encoded prior to scaling: nominal features (e.g., gender, occupation, marital status) are one-hot encoded, while ordinal features (e.g., education level) are mapped to integer codes preserving their natural ordering. For binary and multi-class targets, we apply label encoding to convert string labels to integer codes $(0, 1, ..., K-1)$. Finally, input features are scaled to $[1, 10]$ using MinMaxScaler to ensure positive values.

*Table 7.* Summary of datasets used in our experiments, sorted by number of samples.

| Dataset | Shape | #Classes | Class Distribution | Task | Target Mapping |
|---|---|---|---|---|---|
| IRIS | $(150, 5)$ | 3 | {0: 50, 1: 50, 2: 50} | Flower species classification | {0: Iris-setosa, 1: Iris-versicolor, 2: Iris-virginica} |
| SEEDS | $(210, 8)$ | 3 | {0: 70, 1: 70, 2: 70} | Grain kernel type classification | {0: Kama seed, 1: Rosa seed, 2: Canada seed} |
| HEARTS | $(303, 14)$ | 2 | {0: 138, 1: 165} | Heart disease detection | {0: No significant heart disease, 1: Significant heart disease} |
| ILPD | $(579, 12)$ | 2 | {0: 414, 1: 165} | Liver disease diagnosis | {0: Liver disease, 1:Healthy} |
| TRANSFUSION | $(748, 5)$ | 2 | {0: 570, 1: 178} | Blood donation prediction | {0: Not donating blood, 1: Donating blood} |
| CONTRACEPTIVE | $(1473, 13)$ | 3 | {0: 629, 1: 511, 2: 333} | Contraceptive method classification | {Types of contraception} |
| COMPAS | $(3518, 8)$ | 2 | {0: 1785, 1: 1733} | Recidivism prediction | {0: No recidivist, 1: Recidivist} |
| MAMMOGRAPHY | $(11183, 7)$ | 2 | {0: 10923, 1: 260} | Breast cancer detection | {0: Healthy, 1: Cancer} |
| DEFAULT | $(30000, 27)$ | 2 | {0: 23364, 1: 6636} | Default payment prediction | {0: No, 1: Yes} |
| SKINNONSKIN | $(245057, 4)$ | 2 | {0: 50859, 1: 194198} | Skin pixel segmentation | {0: Non-skin, 1: Skin pixel} |
| LOAN | $(395492, 21)$ | 2 | {0: 355735, 1: 39757} | Loan default risk prediction | {0: Good loan, 1: Bad loan} |

## E.2. Training and Validation Protocol

To ensure robust performance estimates and minimize the risk of overfitting, all classification experiments follow a standardized validation pipeline.

**Data splitting.** We partition each dataset into an initial 80/20 train-test split. The 80% training portion is further utilized within a 5-fold stratified cross-validation (CV) loop for hyperparameter selection. For ECSEL, we additionally reserve an internal 20% validation set from each training fold to manage early stopping and optimization monitoring. Once the optimal parameters are identified via CV, the model is retrained on the full 80% training set and finally evaluated on the held-out 20% test set.

**Optimization and reproducibility.** All experiments are conducted using a fixed random seed (42) to ensure reproducibility. ECSEL is optimized using the Adam optimizer with gradient clipping (`max_norm` = 1.0). For binary classification, we select the best-performing activation between sigmoid and softmax, while for multi-class tasks, we utilize softmax exclusively.

## E.3. Hyperparameter Optimization

Hyperparameter tuning is performed using Optuna's TPE sampler over 30 trials per model. The search ranges, detailed in Table 8, were selected to balance exploration breadth with computational efficiency based on established best practices and preliminary experiments.

For regularization parameters (`C` in Logistic Regression and SVM, `l1_strength` in ECSEL), we use log-uniform distributions spanning 2–4 orders of magnitude to efficiently explore both strong and weak regularization regimes. Tree-based parameters (`max_depth`, `n_estimators`) follow ranges standard in the literature (Probst et al., 2019) that prevent both underfitting (depths too shallow) and overfitting (depths too deep), while keeping computational costs reasonable. Learning rates use log-uniform sampling over $[10^{-4}, 10^{-2}]$ for gradient-based methods, covering typical ranges for Adam optimization. Batch sizes are restricted to powers of two for computational efficiency ($\{32, 64, 128\}$), balancing gradient noise and memory constraints. The number of terms `K` in ECSEL is limited to $[1, 3]$ to maintain interpretability while allowing sufficient model capacity. Early stopping patience values $\{20, 50\}$ were chosen to allow convergence without excessive training time. Finally, the sigmoid threshold range $\{0.4, 0.5, 0.6, 0.7\}$ explores different decision boundaries for

the optional sigmoid transformation in ECSEL.

All other hyperparameters not listed in Table 8 are kept at scikit-learn or PyTorch defaults.

*Table 8.* Hyperparameter search spaces for Optuna optimization (30 trials, TPE sampler).

| Method | Hyperparameter | Search Space | Distribution |
|---|---|---|---|
| Logistic Regression | C 
 max_iter | $[10^{-3}, 10]$ 
 $[100, 1000]$ | Log-uniform 
 Uniform (int) |
| Random Forest | n_estimators 
 max_depth | $[50, 200]$ 
 $[2, 10]$ | Uniform (int) 
 Uniform (int) |
| XGBoost | n_estimators 
 max_depth 
 learning_rate | $[50, 200]$ 
 $[2, 10]$ 
 $[0.01, 0.3]$ | Uniform (int) 
 Uniform (int) 
 Log-uniform |
| SVM | C 
 kernel | $[0.01, 10]$ 
 $\{\texttt{linear}, \texttt{rbf}\}$ | Log-uniform 
 Categorical |
| MLP | hidden_layer_sizes 
 activation | $[10, 100]$ 
 $\{\texttt{relu}, \texttt{tanh}\}$ | Uniform (int, single layer) 
 Categorical |
| ECSEL | K 
 l1_strength 
 batch_size 
 learning_rate 
 num_epochs 
 patience 
 sigmoid_threshold | $[1, 3]$ 
 $[10^{-4}, 10^{-2}]$ 
 $\{32, 64, 128\}$ 
 $[10^{-4}, 10^{-2}]$ 
 $[800, 1000]$ 
 $\{20, 50\}$ 
 $\{0.4, 0.5, 0.6, 0.7\}$ | Uniform (int) 
 Log-uniform 
 Categorical 
 Log-uniform 
 Uniform (int) 
 Categorical 
 Categorical |

### E.4. Full Quantitative Results

This section provides the complete performance data for the classification benchmark. We first present a comparative summary of ECSEL against the strongest baselines followed by the full metric set for all tested configurations.

**Benchmark summary.** Table 9 provides a head-to-head comparison between ECSEL and the top-performing baseline per dataset, reporting accuracy and F1-score with standard deviation over 5-fold cross-validation. Standard deviations for remaining metrics follow similar magnitudes and are omitted for compactness. This summary highlights the competitive parity of ECSEL; across the majority of tasks, ECSEL achieves F1 and Accuracy scores within 1–2% of the best black-box models (often XGBoost) while maintaining a transparent symbolic form.

**Extended benchmark results.** Table 10 provides the results on all metrics, including Accuracy, F1-Score, Precision, and training times. We specifically distinguish between Recall, the overall sensitivity to the positive class defined as $\frac{TP}{TP+FN}$, and Minority Recall, which highlights the sensitivity for the underrepresented class via the ratio $\frac{TP_{min}}{N_{min}}$. This distinction is critical for high-stakes domains with severe class imbalance, such as fraud detection, where the cost of a False Negative in the minority class far outweighs the cost of False Positives.

*Table 9.* Summary of the full classification benchmark. For each dataset, we compare ECSEL against the best-performing baseline by accuracy (ties are broken by F1-score). CV Accuracy and CV F1 report mean $\pm$ std over 5-fold cross-validation; other metrics are test set results.

| Dataset | Method | Accuracy | F1 | Min. Recall | CV Accuracy | CV F1 |
|---|---|---|---|---|---|---|
| IRIS | SVM | **100.00** | 100.00 | – | $97.50 \pm 3.33$ | $97.47 \pm 3.38$ |
| | ECSEL | 96.67 | 96.67 | – | $96.67 \pm 3.12$ | $96.63 \pm 3.16$ |
| SEEDS | LR | 92.86 | 92.77 | – | $96.43 \pm 3.46$ | $96.44 \pm 3.42$ |
| | ECSEL | **97.62** | 97.62 | – | $94.69 \pm 4.31$ | $94.74 \pm 4.22$ |
| HEARTS | LR | 81.97 | 81.54 | 67.86 | $81.82 \pm 4.03$ | $81.74 \pm 4.03$ |
| | ECSEL | **83.61** | 83.61 | 82.14 | $82.70 \pm 6.02$ | $82.64 \pm 6.11$ |
| ILPD | XGBoost | 72.41 | 63.03 | 6.06 | $70.84 \pm 1.54$ | $61.85 \pm 1.38$ |
| | ECSEL | **75.86** | 74.39 | 42.42 | $72.36 \pm 2.44$ | $69.41 \pm 2.92$ |
| TRANSFUSION | XGBoost | **80.67** | 78.72 | 38.89 | $77.26 \pm 0.31$ | $73.46 \pm 1.10$ |
| | ECSEL | 79.33 | 77.95 | 41.67 | $80.26 \pm 1.86$ | $77.04 \pm 2.35$ |
| CONTRACEPTIVE | XGBoost | **60.00** | 59.14 | – | $55.60 \pm 2.66$ | $55.16 \pm 2.40$ |
| | ECSEL | 56.27 | 55.94 | – | $53.73 \pm 3.89$ | $53.75 \pm 3.51$ |
| COMPAS | XGBoost | 68.18 | 68.08 | 62.54 | $68.73 \pm 2.15$ | $68.69 \pm 2.15$ |
| | ECSEL | **68.47** | 68.36 | 62.82 | $68.62 \pm 1.67$ | $68.52 \pm 1.65$ |
| DEFAULT | SVM | **81.82** | 79.35 | 33.61 | $82.00 \pm 0.30$ | $79.58 \pm 0.24$ |
| | ECSEL | 81.74 | 79.34 | 34.06 | $81.81 \pm 0.37$ | $79.17 \pm 0.55$ |
| SKINNONSKIN | XGBoost | **99.96** | 99.96 | 99.97 | $99.95 \pm 0.01$ | $99.95 \pm 0.01$ |
| | ECSEL | 99.25 | 99.25 | 99.88 | $99.47 \pm 0.04$ | $99.47 \pm 0.04$ |
| MAMMOGRAPHY | XGBoost | **98.70** | 98.61 | 59.62 | $98.77 \pm 0.15$ | $98.66 \pm 0.18$ |
| | ECSEL | 98.66 | 98.57 | 59.62 | $98.64 \pm 0.13$ | $98.48 \pm 0.17$ |
| LOAN | XGBoost | **99.51** | 99.50 | 95.55 | $99.50 \pm 0.03$ | $99.50 \pm 0.03$ |
| | ECSEL | 99.22 | 99.21 | 92.97 | $99.21 \pm 0.05$ | $99.20 \pm 0.05$ |

*Table 10.* Full classification benchmark results across all datasets and metrics.

| Dataset | Method | Accuracy | F1 | Precision | Recall | Min. Recall | Training time (s) |
|---|---|---|---|---|---|---|---|
| IRIS (multi) | Logistic Regression | 93.33 | 93.33 | 93.33 | 93.33 | – | 0.002 |
| | Random Forest | 96.67 | 96.67 | 96.67 | 96.67 | – | 0.055 |
| | XGBoost | 96.67 | 96.67 | 96.67 | 96.67 | – | 0.127 |
| | SVM | **100** | **100** | 100 | 100 | – | 0.001 |
| | MLP | 93.33 | 93.33 | 93.33 | 93.33 | – | 0.070 |
| | ECSEL | 96.67 | 96.67 | 96.67 | 96.67 | – | 0.365 |
| SEEDS (multi) | Logistic Regression | 92.86 | 92.77 | 94.12 | 92.86 | – | 0.006 |
| | Random Forest | 90.48 | 90.07 | 91.90 | 90.48 | – | 0.056 |
| | XGBoost | 88.10 | 87.31 | 89.95 | 88.10 | – | 0.494 |
| | SVM | 92.86 | 92.77 | 94.12 | 92.86 | – | 0.001 |
| | MLP | 92.86 | 92.77 | 94.12 | 92.86 | – | 0.105 |
| | ECSEL | **97.62** | **97.62** | 97.78 | 97.62 | – | 0.406 |
| HEARTS (binary) | Logistic Regression | 81.79 | 81.54 | 83.46 | 81.97 | 67.86 | 0.001 |
| | Random Forest | 80.33 | 79.75 | 82.22 | 80.33 | 64.29 | 0.087 |
| | XGBoost | 80.33 | 80.25 | 80.35 | 80.33 | 75.00 | 0.101 |
| | SVM | 81.79 | 81.54 | 83.46 | 81.97 | 67.86 | 0.108 |
| | MLP | 73.77 | 73.51 | 73.90 | 73.77 | 73.77 | 0.113 |
| | ECSEL | **83.61** | **83.61** | 83.61 | 83.61 | **82.14** | 0.144 |
| ILPD (binary) | Logistic Regression | 71.55 | 59.69 | 51.20 | 71.55 | 0.00 | 0.002 |
| | Random Forest | 71.55 | 59.69 | 51.20 | 71.55 | 0.00 | 0.052 |
| | XGBoost | 72.41 | 63.03 | 70.89 | 72.41 | 6.06 | 0.122 |
| | SVM | 71.55 | 59.69 | 51.20 | 71.55 | 0.00 | 0.016 |
| | MLP | 62.57 | 61.29 | 60.64 | 62.07 | 27.27 | 0.144 |
| | ECSEL | **75.86** | **74.39** | 74.25 | 75.86 | **42.42** | 0.272 |
| TRANSFUSION (binary) | Logistic Regression | 78.00 | 71.00 | 78.43 | 78.00 | 11.11 | 0.001 |
| | Random Forest | 80.00 | **78.81** | 78.51 | 80.00 | 44.44 | 0.091 |
| | XGBoost | **80.67** | 78.72 | 79.04 | 80.67 | 38.89 | 0.139 |
| | SVM | 78.00 | 71.82 | 76.67 | 78.00 | 13.89 | 0.028 |
| | MLP | 80.00 | 78.16 | 78.21 | 80.00 | 38.89 | 8.557 |
| | ECSEL | 79.33 | 77.95 | 77.63 | 79.33 | **41.67** | 1.357 |
| CONTRACEPTIVE (multi) | Logistic Regression | 53.90 | 53.03 | 53.27 | 53.90 | – | 0.009 |
| | Random Forest | 54.24 | 53.04 | 53.40 | 54.24 | – | 0.076 |
| | XGBoost | **60.00** | **59.14** | 60.19 | 60.00 | – | 0.186 |
| | SVM | 55.93 | 55.57 | 55.44 | 55.93 | – | 0.151 |
| | MLP | 54.58 | 53.94 | 53.93 | 54.58 | – | 0.364 |
| | ECSEL | 56.27 | 55.94 | 56.12 | 56.27 | – | 4.359 |
| COMPAS (binary) | Logistic Regression | 67.90 | 67.88 | 67.90 | 67.90 | 65.71 | 0.197 |
| | Random Forest | 66.76 | 66.66 | 66.87 | 67.76 | 61.38 | 0.050 |
| | XGBoost | 68.18 | 68.08 | 68.33 | 68.18 | 62.54 | 0.149 |
| | SVM | 65.06 | 65.06 | 65.06 | 65.06 | 65.13 | 0.639 |
| | MLP | 67.76 | 67.76 | 67.76 | 67.76 | **67.44** | 0.280 |
| | ECSEL | **68.47** | **68.36** | 68.62 | 68.47 | 62.82 | 4.773 |
| DEFAULT (binary) | Logistic Regression | 80.92 | 77.29 | 79.05 | 80.92 | 25.24 | 1.880 |
| | Random Forest | 81.60 | 79.32 | 79.75 | 81.60 | 34.66 | 3.628 |
| | XGBoost | 81.73 | **79.43** | 79.94 | 81.73 | 34.66 | 0.465 |
| | SVM | **81.82** | 79.35 | 80.08 | 81.82 | 33.61 | 50.15 |
| | MLP | 81.72 | **79.43** | 79.91 | 81.72 | **34.82** | 1.791 |
| | ECSEL | 81.74 | 79.34 | 79.94 | 81.73 | 34.06 | 56.78 |
| SKINNONSKIN (binary) | Logistic Regression | 91.66 | 91.73 | 91.83 | 91.66 | 82.35 | 0.228 |
| | Random Forest | 99.80 | 99.80 | 99.80 | 99.80 | **99.99** | 5.543 |
| | XGBoost | **99.96** | **99.96** | 99.99 | 99.96 | 99.97 | 0.754 |

Table 10 – continued from previous page

| Dataset | Method | Accuracy | F1 | Precision | Recall | Min. Recall | Training time (s) |
|---|---|---|---|---|---|---|---|
| | SVM | 99.87 | 99.87 | 99.87 | 99.87 | 99.91 | 26.82 |
| | MLP | 99.88 | 99.88 | 99.88 | 99.88 | **99.99** | 59.03 |
| | ECSEL | 99.25 | 99.25 | 99.27 | 99.25 | 99.88 | 108.5 |
| MAMMOGRAPHY (binary) | Logistic Regression | 98.26 | 97.96 | 97.98 | 98.26 | 36.54 | 0.157 |
| | Random Forest | 98.66 | 98.48 | 98.54 | 98.66 | 0.500 | 0.285 |
| | XGBoost | **98.70** | **98.6**1 | 98.59 | 98.70 | **59.62** | 0.217 |
| | SVM | 98.39 | 98.15 | 98.18 | 98.39 | 42.31 | 0.553 |
| | MLP | 98.61 | 98.50 | 98.48 | 98.61 | 55.77 | 11.58 |
| | ECSEL | 98.66 | 98.57 | 98.54 | 98.66 | **59.62** | 10.09 |
| LOAN (binary) | Logistic Regression | 96.16 | 95.95 | 96.00 | 96.16 | 70.18 | 6.778 |
| | Random Forest | 98.74 | 98.73 | 98.73 | 98.47 | 91.01 | 56.49 |
| | XGBoost | **99.51** | **99.50** | 99.51 | 99.51 | **95.55** | 2.394 |
| | SVM[a] | – | – | – | – | – | – |
| | MLP[a] | – | – | – | – | – | – |
| | ECSEL | 99.22 | 99.21 | 98.35 | 99.92 | 92.97 | 102.55 |

[a] SVM and MLP results on LOAN are omitted due to prohibitive training times; MLP required 1049s to fit, and SVM did not converge within the time budget.

### E.5. Comparison Against Matched Per-Class MLPs

To isolate the contribution of ECSEL's signomial parameterization relative to a structurally similar architecture, we compare against per-class MLPs (PCM) with $K \in [1, 3]$ hidden units. This comparison is motivated by the observation that ECSEL can be viewed as a per-class network with exponential activations on log-transformed features: assigning $K$ hidden units per class and aggregating via softmax mirrors ECSEL's structure, with the key difference being the activation function. Matching $K \in [1, 3]$ therefore isolates precisely what the signomial parameterization contributes over a standard piecewise-linear activation. We evaluate two variants: raw features and log-transformed features, the latter directly corresponding to ECSEL's input representation. All other training details follow the same protocol as the main benchmark (Section 5.1): 5-fold stratified CV with Optuna TPE sampler, optimizing over $K \in [1, 3]$, activation $\in \{\texttt{relu}, \texttt{tanh}\}$, learning rate, and batch size.

Table 11 reports the results, with ECSEL achieving higher accuracy and F1 on 10 of 11 datasets. Training times are comparable across methods with no consistent advantage: ECSEL is faster on some datasets (e.g., SEEDS, HEARTS) and slower on others (e.g., LOAN). On two datasets (ILPD and DEFAULT), the per-class MLP performs substantially below ECSEL and the other baselines (see Table 10), with DEFAULT showing near-complete failure (22% accuracy, F1 = 8.01 for the raw variant). A likely explanation is that with ReLU activations and only 1–3 neurons per class, neurons with persistently negative pre-activations output zero and stop receiving gradient updates. ECSEL's signomial terms, by contrast, always produce a non-zero output for positive inputs since each term $\alpha_k \prod_j x_j^{\beta_{k,j}} \neq 0$, ensuring that gradients flow through every term during training. We note that Optuna selected ReLU over tanh on these datasets despite both being available.

Beyond raw performance, the two architectures differ in the type of interpretability they afford. Signomials offer a different kind of transparency than neural networks: the elasticity interpretation of each $\beta_{k,j}$ holds globally across the input space, whereas the effective contribution of a neuron's weights in the per-class MLP depends on which region of the input space activates that neuron. To illustrate, consider $K=2$ with 4 features. The per-class MLP computes

$$z_0 = v_1 \sigma(w_{11}x_1 + \cdots + b_1) + v_2 \sigma(w_{21}x_1 + \cdots + b_2) + b_0,$$

where $v_k$, $w_{ki}$, and $b_k$ are learned weights and biases, and the activations introduce input-dependent gating: each neuron's weights are only active when the neuron fires, making feature effects region-dependent. For $K = 1$ this reduces to a gated linear model (comparable to logistic regression in the active region), but for $K > 1$ the neurons activate in different (or overlapping) regions of the input space, making the resulting feature weights region-dependent and difficult to summarize globally. ECSEL computes

$$z_0 = \alpha_1 x_1^{\beta_{1,1}} x_2^{\beta_{1,2}} x_3^{\beta_{1,3}} x_4^{\beta_{1,4}} + \alpha_2 x_1^{\beta_{2,1}} x_2^{\beta_{2,2}} x_3^{\beta_{2,3}} x_4^{\beta_{2,4}},$$

where each $\beta_{k,j}$ is the elasticity of term $k$ with respect to feature $j$: a 1% increase in $x_j$ changes the term's contribution

by $\beta_{k,j}\%$, and this holds globally across the input space. This structure directly yields closed-form elasticities (G1), exact counterfactuals (G2), and additive log-space decompositions (L1) from Section 3.2, none of which follow from a piecewise-linear parameterization.

In practice, explaining the per-class MLP requires additional computation. Raw gradients (one backward pass) provide local sensitivity but not additive attribution. Global importance requires aggregation over the dataset, and proper additive decompositions require methods such as Integrated Gradients or SHAP. ECSEL provides all three directly from the learned expression.

*Table 11.* Performance comparison of ECSEL against matched per-class MLPs (PCMs) with raw and log-transformed features.

| | ECSEL | | | PCM (raw) | | | PCM (log) | | |
|---|---|---|---|---|---|---|---|---|---|
| **Dataset** | Accuracy | F1 | Training time (s) | Accuracy | F1 | Training time (s) | Accuracy | F1 | Training time (s) |
| IRIS | **96.67** | **96.67** | 0.4 | 96.67 | 96.67 | 0.5 | 93.33 | 93.33 | 0.5 |
| SEEDS | **97.62** | **97.62** | 0.1 | 92.86 | 92.77 | 0.8 | 88.10 | 87.31 | 0.7 |
| HEARTS | **83.61** | **83.61** | 0.1 | 80.33 | 79.97 | 0.2 | **83.61** | 83.44 | 0.7 |
| ILPD | **75.86** | **74.39** | 0.3 | 58.62 | 60.33 | 0.5 | 58.62 | 59.43 | 0.6 |
| TRANSFUSION | **79.33** | 77.95 | 1.4 | 78.00 | **78.63** | 1.4 | 76.67 | 78.04 | 1.6 |
| CONTRACEPTIVE | **56.27** | **55.94** | 4.4 | 53.22 | 53.57 | 1.2 | 53.90 | 54.73 | 0.4 |
| COMPAS | **68.47** | **68.36** | 4.8 | 67.33 | 67.32 | 1.2 | 68.04 | 67.94 | 0.8 |
| DEFAULT | **81.74** | **79.34** | 56.8 | 22.12 | 8.01 | 2.5 | 74.97 | 76.14 | 13.2 |
| SKINNONSKIN | 99.25 | 99.25 | 108.5 | 99.86 | 99.86 | 118.1 | **99.88** | **99.88** | 147.7 |
| MAMMOGRAPHY | **98.66** | **98.57** | 10.1 | 94.14 | 95.65 | 7.6 | 95.71 | 96.61 | 11.6 |
| LOAN | **99.22** | **99.21** | 102.6 | 98.78 | 98.79 | 93.2 | 99.12 | 99.12 | 291.1 |

# F. Case Study: Online Shopping Purchase Intent

Converting website visitors into paying customers remains a central challenge in e-commerce optimization. Predicting purchase intent enables e-commerce platforms to deploy targeted interventions such as personalized recommendations and dynamic pricing.

## F.1. Dataset

The Online Shoppers Intention dataset captures user interactions on an e-commerce website, with 12,330 sessions over a time span of a year, and the binary target variable *Revenue* indicating whether a purchase occurred. It consists of 18 attributes reflecting a user's characteristics such as number of page visits or its browser type. Overall, 1,908 sessions resulted in purchases (15.5%), highlighting a class imbalance that informs subsequent preprocessing and modeling decisions. The dataset's features as described by Sakar et al. (2018) can be found in Table 12.

*Table 12.* Features of the Online Shopping Intent dataset

| Feature name | Feature description | Min. value | Max. value | SD / # Categories |
|---|---|---|---|---|
| **Numerical features** | | | | |
| Administrative | Number of pages visited by the visitor about account management | 0 | 27 | 3.32 |
| Administrative duration | Total amount of time (in seconds) spent by the visitor on account management related pages | 0 | 3398 | 176.70 |
| Informational | Number of pages visited by the visitor about Web site, communication and address information of the shopping site | 0 | 24 | 1.26 |
| Informational duration | Total amount of time (in seconds) spent by the visitor on informational pages | 0 | 2549 | 140.64 |
| Product related | Number of pages visited by visitor about product related pages | 0 | 705 | 44.45 |
| Product related duration | Total amount of time (in seconds) spent by the visitor on product related pages | 0 | 63,973 | 1912.25 |
| Bounce rate | Average bounce rate value of the pages visited by the visitor | 0 | 0.2 | 0.04 |
| Exit rate | Average exit rate value of the pages visited by the visitor | 0 | 0.2 | 0.05 |
| Page value | Average page value of the pages visited by the visitor | 0 | 361 | 18.55 |
| Special day | Closeness of the site visiting time to a special day | 0 | 1.0 | 0.19 |
| **Categorical features** | | | | |
| OperatingSystems | Operating system of the visitor | – | | 8 |
| Browser | Browser of the visitor | – | | 13 |
| Region | Geographic region from which the session has been started by the visitor | – | | 9 |
| TrafficType | Traffic source by which the visitor has arrived at the Web site (e.g., banner, SMS, direct) | – | | 20 |
| VisitorType | Visitor type as "New Visitor," "Returning Visitor," and "Other" | – | | 3 |
| Weekend | Boolean value indicating whether the date of the visit is weekend | – | | 2 |
| Month | Month value of the visit date | – | | 12 |
| Revenue | Class label indicating whether the visit has been finalized with a transaction (target) | – | | 2 |

## F.2. Preprocessing and Feature Engineering

To improve model interpretability and predictive performance, we apply a feature engineering consisting of redundancy removal, categorical aggregation, and domain-informed feature construction. This process reduces the original 18 features to 14 more informative variables while preserving essential behavioral signals.

**Redundancy Removal.**   Correlation analysis revealed strong dependencies between page count and duration features: *ProductRelated* and *ProductRelated_Duration* (0.86), *Informational* and *Informational_Duration* (0.62), and *Administrative* and *Administrative_Duration* (0.60). Mutual information analysis confirmed that duration features provide no additional predictive signal beyond their count counterparts ($\text{MI}_{\text{count}} \geq \text{MI}_{\text{duration}}$).

*BounceRates* and *ExitRates* exhibited high correlation while measuring similar engagement patterns. However, *ExitRates* demonstrated significantly higher mutual information with the target (MI = 0.042 vs. 0.025) and was selected earlier by the mRMR algorithm, indicating superior relevance with lower redundancy. Consequently, *BounceRates* was removed in favor of *ExitRates*.

**Categorical Aggregation.** Several categorical features exhibited highly skewed distributions with many low-frequency categories. *TrafficType* (20 categories) was aggregated into the top three traffic sources plus an "Other" category. *Browser* (13 categories) was similarly reduced to the top two browsers plus "Other". *OperatingSystems* (8 categories) was binarized into mobile versus desktop. *VisitorType* was converted to a binary indicator distinguishing returning visitors from new and other visitor types. *Month* was encoded numerically to preserve temporal ordering.

**Feature Engineering.** To capture nonlinear interactions and domain-specific patterns, we constructed three engineered features:

- *ProductFocusRatio.* The proportion of product-related pages among all pages visited, defined as

$$\frac{ProductRelated}{ProductRelated + Administrative + Informational}$$

Values near 1 indicate focused product browsing rather than general site exploration.

- *PageValue_per_ExitRate (PVER).* The ratio of *PageValues* to *ExitRates*, capturing the intuition that high-value pages with low exit rates signal strong purchase intent. To prevent division-by-zero and stabilize extreme values, the feature is clipped to $[0, 1000]$.

- *ShopIntensity.* A composite engagement metric combining *ProductRelated* (weight 0.3), *PageValues* (weight 0.5), and *ProductFocusRatio* (weight 0.2), normalized to a $[0, 10]$ scale. This feature aggregates multiple behavioral signals into a single intensity measure.

The final preprocessed dataset comprises 14 features: 6 original numeric features (*ProductRelated*, *Administrative*, *Informational*, *ExitRates*, *PageValues*, *SpecialDay*), 1 temporal feature (*Month*), 4 aggregated categorical features (*TrafficType_Grouped*, *Browser_Grouped*, *IsMobile*, *IsReturning*), and 3 engineered features (*ProductFocusRatio*, *PageValue_per_ExitRate* and *ShoppingIntensity*).

### F.3. Experimental Setup

**Training Details.** The dataset was split into 70% training and 30% test sets, consistent with prior work on this dataset ((Sakar et al., 2018), (Baati & Mohsil, 2020)). Hyperparameter optimization was performed using Optuna's TPE sampler with 5-fold stratified cross-validation on the training set to account for class imbalance. All features were scaled to $[1, 10]$ using MinMax scaling to maintain numerical stability during gradient-based optimization of the signomial expressions. ECSEL was trained using the Adam optimizer with early stopping based on validation loss. The search space and optimal hyperparameters are presented in Table 13.

**Hyperparameter Search Space.** The search spaces, detailed in Table 13, balance exploration breadth with computational constraints but prioritize interpretability by fixing $K = 1$ such that ECSEL could learn a single-term signomial classifier.

*Table 13.* Hyperparameter search space and optimal values from Optuna optimization

| Hyperparameter | Search Space | Optimal Value |
|---|---|---|
| $K$ | $\{1\}$ | 1 |
| $\ell_1$ | $[10^{-4}, 10^{-2}]$ (log) | 0.0086 |
| Class Weight Multiplier | $[0.01, 1.0]$, step 0.1 | 0.91 |
| Learning Rate | $[10^{-5}, 10^{-2}]$ (log) | $3.39 \times 10^{-4}$ |
| Batch Size | $\{128, 256\}$ | 256 |
| Num Epochs | $[100, 200]$, step 25 | 150 |
| Patience | $[25, 70]$ | 47 |
| Num Restarts | $\{1\}$ | 1 |
| **Best CV F1 Score: 0.6891** | | |

## F.4. Results

As reported in Section 6.1, the signomial that reports the best performance is given by

$$z = 0.10 \cdot \frac{PageValues^{0.47} \cdot Month^{0.07} \cdot PageValue\_per\_ExitRate^{1.09} \cdot ShopIntensity^{0.66}}{ExitRates^{0.41} \cdot Administrative^{0.14} \cdot IsReturn^{0.04}}$$

**Comparison to Baselines.** Table 14 presents the full test performance of ECSEL against all baseline classifiers. XGBoost achieves the highest accuracy (89.86%) and F1-score (67.02%), while SVM leads on minority recall (76.57%), closely followed by ECSEL (75.70%) and MLP (75.35%). Notably, the decision threshold plays an important role here: SVM, MLP, and ECSEL all operate at non-default thresholds (0.247, 0.304, and 0.559 respectively), reflecting the dataset's class imbalance. Among the methods competitive on minority recall, ECSEL is the only one that produces a closed-form, directly inspectable expression. In terms of computational efficiency, ECSEL's training time (5.5s) is practical, sitting between the fast tree-based methods (0.2–3.0s) and Logistic Regression (8.0s).

*Table 14.* Test performance comparison of classification models on the Online Shopping Intention dataset.

| Model | Accuracy (%) | F1 (%) | Recall (%) | Training time (s) |
|---|---|---|---|---|
| LR | 87.02 | 63.75 | 73.78 | 8.0 |
| RF | 88.83 | 65.38 | 68.18 | 0.3 |
| XGBoost | **89.86** | **67.02** | 66.61 | 0.2 |
| SVM | 87.08 | 64.70 | 76.57 | 3.0 |
| MLP | 87.08 | 64.33 | 75.35 | 1.0 |
| ECSEL | 87.10 | 64.48 | 75.70 | 5.5 |

**Regularization Trade-off Study.** To investigate the trade-off between model sparsity and predictive performance, we trained a second model with reduced $\ell_1$ regularization strength ($\lambda' = 0.00135$ versus $\lambda = 0.00864$). This weaker regularization yielded a longer, more complex formula retaining 14 features:

$$z = 0.07 \cdot \frac{PageValues^{0.63} \cdot Month^{0.14} \cdot IsReturning^{0.07} \cdot PVER^{1.05} \cdot ShopIntensity^{0.79}}{ProductRelated^{0.10} \cdot ExitRates^{0.62} \cdot Administrative^{0.16} \cdot Info^{0.15} \cdot SpecialDay^{0.13} \cdot ProductFocusRatio^{0.04}} \quad \text{(A10)}$$

Table 15 compares both formulations. The dense model achieves marginally lower test performance (F1 = 0.644 vs. 0.646, ROC-AUC = 0.877 vs. 0.880) despite using twice as many features, demonstrating that additional complexity provides negligible predictive benefit.

Both models identify *PageValue_per_ExitRate* as the dominant predictor ($\beta = 1.09$ sparse, 1.05 dense), with *ShopIntensity* and *PageValues* showing strong positive effects and *ExitRates* inversely related. These consistent patterns across regularization strengths confirm the robustness of these purchase drivers.

The dense model, however, retains several weak predictors that likely reflect noise rather than signal. *ProductRelated* ($\beta = -0.10$), *Informational* ($-0.15$), and *SpecialDay* ($-0.13$) show small negative effects that may capture browsing without purchase intent. *ProductFocusRatio* ($-0.04$) adds little beyond *ShopIntensity*, illustrating redundancy among engineered features. Minor sign reversals for *IsReturning* between models ($|\beta| \leq 0.07$) underscore the instability of weak predictors. The sparse model's stronger regularization correctly identifies and eliminates such uninformative features, yielding a more parsimonious expression with equivalent or superior performance.

## F.5. Full Interpretability Analysis

**Inspection of the Signomial.** The learned formula reveals clear purchase drivers. *PageValue_per_ExitRate* ($\beta = 1.09$) is the dominant predictor, confirming that high-value pages with low exit rates signal strong intent. This engineered feature outperforms its constituents, validating domain-informed feature design. *ShopIntensity* ($\beta = 0.66$) and *PageValues* ($\beta = 0.47$) reinforce that engagement quality exceeds browsing quantity as a conversion predictor.

Negative effects include *ExitRates* ($\beta = -0.41$) and *Administrative* pages ($\beta = -0.14$), indicating that frequent exits and help-seeking reduce purchase likelihood. The weak negative *IsReturning* effect ($\beta = -0.04$) may reflect deliberation

*Table 15.* Performance comparison of sparse versus dense signomial formulations

| Metric | Equation 15 ($\lambda_1 = 0.00864$) | Equation A10 ($\lambda_1 = 0.00135$) |
|---|---|---|
| Active Features | 7 | 14 |
| Test Accuracy | **0.872** | 0.871 |
| Test F1 Score | **0.646** | 0.644 |
| Test Recall | 0.757 | **0.759** |
| Test Precision | **0.563** | 0.560 |
| Test ROC-AUC | **0.880** | 0.877 |
| CV F1 Score | **0.689** | 0.688 |
| Training Time (s) | **4.4** | 5.1 |

behavior among returning non-purchasers. *Month* ($\beta = 0.07$) captures seasonal patterns, with later months showing higher conversion, something that also shows in the distribution of this feature.

$\ell 1$ regularization eliminated low-impact categorical features (traffic source, browser, device), yielding a sparse seven-feature model. The multiplicative signomial structure naturally encodes that purchase requires simultaneous satisfaction of multiple engagement conditions, providing both predictive accuracy and interpretable business insights.

**Explanations through Desirable Properties.** Figure 4 demonstrates ECSEL's desirable properties (Section 3.2) through four complementary visualizations applied to representative test instances, each reflecting the learned formula's power-law behavior.

*Counterfactual scaling (Figure 4a)* demonstrates exact counterfactual predictions (Property G2) by showing how proportional changes to individual features affect purchase probability. We select Instance 6457, a true negative correctly predicted with $p = 0.555$ (just below the decision threshold of $p = 0.559$), as the baseline for this analysis. Increasing *ShopIntensity* raises the probability monotonically, while increasing *ExitRates* or *Administrative* reduces it, consistent with their learned elasticities. All curves intersect at the baseline ($q = 1$), confirming ECSEL's multiplicative structure enables exact counterfactual reasoning.

*Scenario comparison (Figure 4b)* evaluates four hypothetical user profiles to illustrate how different behavioral patterns translate into predictions. The *Average user* profile, constructed from feature means, yields a probability below the threshold ($p < 0.559$), reflecting the dataset's default no-purchase outcome (15.5% base conversion rate). Two negative scenarios, *High ExitRates* and *High Administrative* (users browsing administrative or help pages extensively), produce strongly negative logits, confirming these as indicators of no-purchase intent. Conversely, the *High PageValues* scenario generates a positive logit that crosses the decision threshold, demonstrating how high-value page engagement drives conversion predictions.

*Instance-level explanations (Figures 4c and 4d)* provide exact local decompositions (Property L1) for two contrasting sessions. Instance 10894 (predicted no-purchase, $p = 0.450$) lies just below the decision threshold, where positive contributions from *PageValue_per_ExitRate* (+0.42) and *ShopIntensity* (+0.18) are insufficient to overcome the negative baseline bias (-0.82) reflecting the default no-purchase outcome. In contrast, Instance 3664 (predicted purchase, $p = 0.684$) exceeds the threshold due to a substantially stronger *PageValue_per_ExitRate* contribution (+0.89), demonstrating how this dominant feature drives conversion predictions when sufficiently elevated. The dashed vertical line in both waterfall plots marks the decision threshold ($p = 0.559$), providing immediate visual feedback on prediction confidence.

Together, these visualizations demonstrate ECSEL's ability to provide exact, actionable explanations not always available in black-box ensemble methods.

**Global feature attribution.** A summary of explanation methods and top-3 features is provided in Table 2 in the main text; Table 16 reports all Spearman rank correlations between ECSEL's G1 feature importance rankings and those produced by SHAP and LIME for each baseline. ECSEL's rankings are strongest for tree-based methods ($\rho = 0.87$ with RF-SHAP, $\rho = 0.80$ with XGBoost-SHAP, $p < 0.001$), and weaker for the MLP ($\rho = 0.23$), which does not reach statistical significance. The weaker MLP agreement is consistent with the structural mismatch between piecewise-linear activations and ECSEL's multiplicative power-law representation, though all nonlinear methods agree on *PVER* as the top predictor.

**G3 vs. LIME.** Since property G3 and LIME both provide local feature attributions, we compare them directly to validate that ECSEL's closed-form sensitivity analysis produces explanations consistent with the established sampling-based

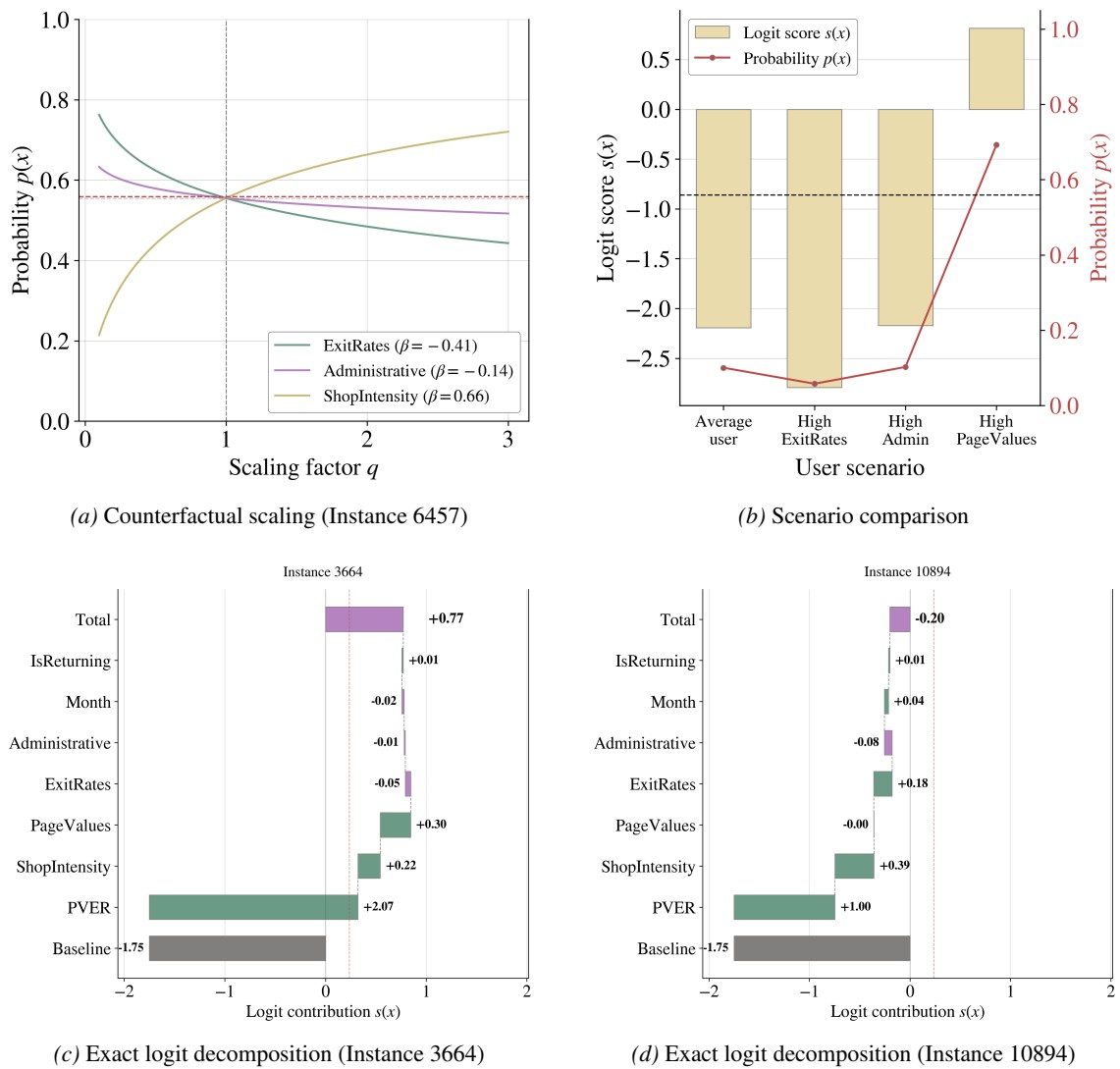

*(a)* Counterfactual scaling (Instance 6457)

*(b)* Scenario comparison

*(c)* Exact logit decomposition (Instance 3664)

*(d)* Exact logit decomposition (Instance 10894)

*Figure 4.* Visual explanations demonstrating ECSEL's desirable properties on the Online Shopping Intention dataset.

*Table 16.* Spearman rank correlation ($\rho$) between ECSEL's G1 feature importance and global feature importance from SHAP and LIME over the full OSI test set. $p$-values indicate statistical significance.

| Model | SHAP | | LIME | |
|---|---|---|---|---|
| | $\rho$ | $p$-val | $\rho$ | $p$-val |
| Logistic Regression | 0.641 | 0.0136 | 0.641 | 0.0136 |
| Random Forest | **0.871** | **0.0001** | 0.711 | 0.0044 |
| XGBoost | **0.796** | **0.0007** | 0.735 | 0.0028 |
| MLP | 0.228 | 0.4338 | 0.181 | 0.5364 |

approach. G3 gives exact, closed-form local sensitivity through the log-gradient, whereas LIME fits a local surrogate via sampling. G3 achieves a mean Spearman $\rho = 0.997$ (median 1.0, std 0.019) with LIME across 200 test instances, with 100% sign agreement on all major features and $\sim 11{,}000\times$ lower computation cost ($0.3\,\mu$s vs $3.2\,$ms per instance). The strong agreement holds because LIME's surrogate is here applied directly to ECSEL, implicitly mirroring its signomial structure in the local region. When LIME is applied to black-box models whose structure differs from a signomial, agreement with G3 naturally weakens, as seen in Table 16 for MLPs. Table 17 shows attributions for two representative instances: no-purchase

instance 3523 ($p = 0.524$, just below the decision threshold) and purchase instance 811 ($p = 0.746$). Both methods agree on the sign and rank of all features, with *PVER*, *ShopIntensity*, and *PageValues* driving purchase predictions and *ExitRates* and *Administrative* working against it.

*Table 17.* G3 vs. LIME local attributions for two representative OSI test instances. Checkmarks indicate sign agreement between G3 and LIME.

| | **No purchase** ($p = 0.524$) | | | **Purchase** ($p = 0.746$) | | |
|---|---|---|---|---|---|---|
| **Feature** | G3 | LIME | | G3 | LIME | |
| *PVER* | $+0.026$ | $+0.076$ | ✓ | $+0.223$ | $+0.074$ | ✓ |
| *ShopIntensity* | $+0.016$ | $+0.016$ | ✓ | $+0.135$ | $+0.021$ | ✓ |
| *PageValues* | $+0.011$ | $+0.011$ | ✓ | $+0.096$ | $+0.016$ | ✓ |
| *ExitRates* | $-0.010$ | $-0.068$ | ✓ | $-0.084$ | $-0.069$ | ✓ |
| *Administrative* | $-0.003$ | $-0.029$ | ✓ | $-0.029$ | $-0.039$ | ✓ |
| *Month* | $+0.002$ | $+0.004$ | ✓ | $+0.014$ | $+0.006$ | ✓ |
| *IsReturning* | $-0.001$ | $-0.001$ | ✓ | $-0.008$ | $-0.002$ | ✓ |

Beyond speed, LIME's sampling procedure introduces seed-dependent variance that G3 avoids entirely. Tan et al. (2023) show theoretically that standard LIME requires exponentially more samples than necessary for convergence, resulting in explanations that vary substantially across seeds. Our experiments corroborate this: on a near-boundary instance ($p \approx 0.50$), LIME rankings for MLP are inconsistent across five random seeds (mean pairwise $\rho = 0.785$). Individual seed correlations against ECSEL's global feature ranking (property G1) range from $0.46$ to $0.66$, two of which do not reach statistical significance ($p > 0.05$), suggesting that LIME's explanation for those seeds is difficult to distinguish from a random ranking. ECSEL avoids this instability by construction, as G3 attributions are derived analytically from the model parameters and require no sampling.

# G. Case Study: PaySim Fraud Detection

The PaySim dataset is a synthetic financial log generated by Lopez-Rojas et al. (2016) to model mobile money transactions, derived from a real African mobile money service. The dataset is available on Kaggle[2]. In this case study, we evaluate ECSEL's performance on fraud detection and provide a detailed comparison with Deep Symbolic Classification (DSC) (Visbeek et al., 2024), which previously applied symbolic regression to this dataset.

## G.1. Data

The dataset consists of approximately 6.3 million transactions generated over 744 time steps, which represents a 30-day simulation period. The target variable is *isFraud* and these fraudulent transactions only make up 0.13% of the dataset (8,213 fraud cases). The dataset's features include:

- *step*: timestep (1 hour)

- *type*: transaction type (cash_in, cash_out, debit, payment, transfer)

- *amount*: transaction amount

- *nameOrig*, *nameDest*: originator and recipient identifiers

- *oldbalanceOrg*, *newbalanceOrig*: originator balances

- *oldbalanceDest*, *newbalanceDest*: recipient balances

- *isFlaggedFraud*: simulator flag

- *isFraud*: fraud label (target)

## G.2. Preprocessing and Feature Engineering

The PaySim dataset requires careful preprocessing to address simulator artifacts, extreme cardinality in categorical features, and redundant or uninformative variables.

**Transaction Type Encoding.**  The categorical *type* feature, which takes five values (cash_in, cash_out, transfer, debit, and payment), is one-hot encoded into four binary indicator variables: *type_CASH_OUT*, *type_TRANSFER*, *type_DEBIT*, and *type_PAYMENT*. Transactions of type cash_in are omitted as the reference category to avoid multicollinearity. This encoding is particularly important because fraud appears to exclusively occur in cash_out and transfer transactions. The binary indicators allow the model to learn type-specific fraud patterns while maintaining interpretability through explicit feature coefficients.

**Excluded Features.**  We deliberately exclude several features that either introduce overfitting risk, contain simulation artifacts, or provide redundant information:

- *nameOrig* and *nameDest* (account identifiers).  These string identifiers exhibit extreme cardinality, with 99.85% unique values for originating accounts and 42.79% for destination accounts across the 6.3M transactions. Direct inclusion would lead to severe overfitting, as the model would memorize account-specific patterns rather than learning generalizable fraud indicators. Instead, we extract structural information through the account type prefix (C for customer accounts, M for merchant accounts) and discard the raw identifiers. This preserves the distinction between customer and merchant while preventing memorization of individual account behavior.

- *step* (raw timestep). The raw timestep variable, representing simulation time in hourly increments, introduces artificial periodicity artifacts inherent to the simulation design rather than reflecting real-world temporal patterns. We replace this with *hour_of_day* (derived via modulo 24 operation), which captures diurnal patterns in transaction activity without encoding the simulation's artificial monthly cycles. This transformation maintains temporal information relevant to fraud detection (e.g., late-night transactions) while removing simulation-specific noise.

---

[2]https://www.kaggle.com/datasets/ealaxi/paysim1

- *isFlaggedFraud (simulator's fraud flag).* This binary indicator, intended to represent a simple rule-based fraud detection system within the simulator (Lopez-Rojas et al., 2016), flags only 16 transactions out of 6.3 million. Its detection rule is deterministic: it flags only TRANSFER transactions exceeding \$200,000. Because this feature is a strict function of *type* and *amount*, both already present in the model, it provides no independent predictive signal and would introduce perfect multicollinearity. Moreover, its near-zero variance (0.0003% positive rate) makes it numerically unstable during optimization.

- *oldbalanceDest* and *newbalanceDest* (destination balances). We exclude destination account balance features based on both data quality concerns and domain considerations. The PaySim documentation explicitly notes that merchant accounts (M-prefixed destinations) consistently report zero balances regardless of transaction flow, reflecting a simulator design choice rather than realistic account behavior. Including these features would allow the model to trivially distinguish customer-to-customer from customer-to-merchant transactions through balance patterns alone, rather than learning meaningful fraud indicators. We preserve account type information through the derived *externalDest* binary feature, which explicitly indicates whether the destination is a merchant account, thereby capturing this distinction without relying on potentially misleading balance values.

**Feature Engineering.** We engineer three additional features to capture transaction patterns indicative of fraudulent behavior:

- *hour_of_day.* The hour of day extracted from the simulation timestep, defined as

$$hour\_of\_day = step \bmod 24$$

This feature captures time-of-day patterns in transaction activity (e.g., late-night transactions) while removing the simulation's artificial monthly cycles present in the raw *step* variable. Values range from 0 to 23, representing the hour within a 24-hour period.

- *pct_balance_taken.* The proportion of the originator's account balance consumed by the transaction, defined as

$$pct\_balance\_taken = \begin{cases} \min\left(\frac{amount}{oldbalanceOrg}, 1\right) & \text{if } oldbalanceOrg > 0 \\ 0 & \text{otherwise} \end{cases}$$

Values near 1 indicate that the transaction drains the account, a pattern commonly associated with account takeover fraud. The feature is clipped to $[0, 1]$ to handle cases where the transaction amount exceeds the recorded balance.

- *externalOrig.* A binary indicator identifying potential external or pass-through accounts, defined as

$$externalOrig = \mathbb{1}[oldbalanceOrg = 0 \text{ and } newbalanceOrig = 0]$$

This feature flags transactions where both pre- and post-transaction balances are zero, capturing accounts at counterparty banks whose balance information is unavailable in the PaySim simulation. Following Visbeek et al. (2024), we construct this indicator alongside *externalDest* (defined analogously for destination accounts) to identify transactions involving external financial institutions.

The final feature set consists of 13 features: 6 original continuous features, 5 one-hot encoded transaction types, and 2 engineered features. Table 18 provides descriptions of all features used in the model.

**Feature Scaling and Transformation.** To ensure numerical stability in the learned signomial expressions while preserving the informative spread of monetary values, we apply a multi-stage scaling pipeline. First, we apply log transformation to monetary features with wide value ranges using $X_{\log} = \ln(X + \epsilon)$ where $\epsilon = 10^{-6}$ handles zero values. Specifically, we transform *amount*, *oldbalanceOrg*, and *newbalanceOrig*.

After train-test splitting (85/15 stratified by fraud rate), we apply StandardScaler to all continuous features (*amount*, *oldbalanceOrg*, *newbalanceOrig*, *hour_of_day*). Consistent with Visbeek et al. (2024), we use StandardScaler rather than MinMax scaling because the extreme range of transaction amounts (ranging across several orders of magnitude even after log transformation) would be excessively compressed into a narrow interval like $[0, 1]$, losing meaningful information between

small and large transactions. StandardScaler preserves the relative spread of values while centering the distribution, allowing the model to learn appropriate scaling factors through the signomial coefficients. The scaler is fit only on the training set to prevent data leakage. Binary and categorical features are preserved in their original $\{0, 1\}$ range without transformation.

The complete preprocessing pipeline follows this order: (1) feature engineering on raw data, (2) feature selection retaining 11 features (7 original, 4 engineered), (3) log transformation of monetary variables, (4) stratified train-test split, (5) standardization of continuous features, (6) preservation of binary features.

*Table 18.* Features used in the PaySim fraud detection experiments. Continuous features are scaled using StandardScaler.

| Feature name | Feature description | Type | Scaled | Notes |
|---|---|---|---|---|
| **Original numerical features** | | | | |
| *step* | Discrete time step of the transaction (1 hour per step) | Continuous | Yes | Transaction timestamp |
| *amount* | Transaction amount | Continuous | Yes | Log-transformed |
| *oldbalanceOrg* | Origin account balance before transaction | Continuous | Yes | Log-transformed |
| *newbalanceOrig* | Origin account balance after transaction | Continuous | Yes | Log-transformed |
| *oldbalanceDest* | Destination account balance before transaction | Continuous | Yes | Log-transformed |
| *newbalanceDest* | Destination account balance after transaction | Continuous | Yes | Log-transformed |
| **Transaction type indicators (one-hot encoded)** | | | | |
| *CashIn (CI)* | Cash-in transaction indicator | Binary | No | One-hot encoded |
| *CashOut (CO)* | Cash-out transaction indicator | Binary | No | One-hot encoded |
| *Debit (D)* | Debit transaction indicator | Binary | No | One-hot encoded |
| *Payment (P)* | Payment transaction indicator | Binary | No | One-hot encoded |
| *Transfer (T)* | Transfer transaction indicator | Binary | No | One-hot encoded |
| **Engineered features** | | | | |
| *pct_balance_taken (PctBT)* | Fraction of the origin balance transferred in the transaction | Continuous | Yes | Clipped to $[0, 1]$ |
| *externalOrig* | Indicator that the origin account is external (both balances equal zero) | Binary | No | Transaction-level rule |

## G.3. Experimental Setup

**Training Details.** We train ECSEL using the Adam optimizer with early stopping based on validation loss. The dataset is split 85/15 into training and test sets using stratified sampling to preserve the fraud rate (0.13% positive class). Hyperparameter optimization is performed using Optuna's TPE sampler with 5-fold stratified cross-validation on the training set. To account for extreme class imbalance, we tune a class weight multiplier that upweights the minority class during training. All experiments are conducted with a fixed random seed (42) for reproducibility.

**Hyperparameters Search Space.** For hyperparameter selection, the 6.3 million-sample dataset was subsampled to 1 million instances via stratified random sampling, maintaining the original fraud prevalence of 0.013%. Given the dataset's scale, we conduct hyperparameter optimization with 5-fold cross-validation and 10 Optuna trials (compared to 30 trials on all other experiments) to balance computational feasibility with adequate search space exploration.

*Table 19.* Hyperparameter search space and optimal values from Optuna optimization

| Hyperparameter | Search Space | Optimal Value |
|---|---|---|
| $K$ | $\{1, 2\}$ | 1 |
| $\ell_1$ | $[10^{-4}, 10^{-2}]$ (log) | $2.05 \times 10^{-4}$ |
| Class Weight Multiplier | $[0.3, 1.0]$, step 0.1 | 0.40 |
| Learning Rate | $[10^{-4}, 10^{-2}]$ (log) | $2.16 \times 10^{-4}$ |
| Batch Size | $\{512, 1024\}$ | 1024 |
| Num Epochs | $[150, 250]$, step 25 | 250 |
| Patience | $[25]$ | 25 |
| Num Restarts | $\{1\}$ | 1 |
| **Best CV F1 Score: 74.48** | | |

## G.4. Results

As given in Section 6.2, the signomial learned by ECSEL is given by:

$$z = -0.07 \cdot \frac{A^{0.02} \cdot P^{0.03}}{exO^{0.03} \cdot exD^{0.16} \cdot CO^{0.14} \cdot T^{0.06} D^{0.03}} + 0.09 \cdot \frac{OBO^{1.42}}{NBO^{0.04} \cdot exD^{0.07} \cdot CO^{0.06} \cdot D^{0.06} P^{0.06}} \cdot \tag{A11}$$

where $A$ is the transaction *amount*, *exO* and exD indicate whether the origin or destination accounts are external (*externalOrig*, *externalDest*), and *OBO* and *NBO* are the origin account balances before and after transaction (*oldbalanceOrig* and *newbalanceOrg*). The transaction types *Transfer*, *Debit*, *Payment* and *CashOut* are denoted by *T, D, P* and *CO* respectively. All features can be found in Table 18 (Appendix G).

On the full test set, ECSEL achieves an F1 score of 79.08%, with minority class recall of 68.10% and precision of 94.27%. These metrics are computed at the optimal classification threshold of $p = 0.904$, substantially higher than the default 0.5 threshold. This elevated threshold reflects the extreme class imbalance (0.13% fraud rate): the model must be highly confident before classifying a transaction as fraudulent to maintain an acceptable precision-recall balance. ECSEL took approximately 16 minutes to train on the full 6.3 million transaction dataset. All optimal parameters can be found in Table 19.

*Table 20.* Test performance comparison of classification baselines on the PaySim fraud detection dataset.

| Method | F1 (%) | Recall (%) | Precision (%) | ROC-AUC | Threshold | Time (s) |
|---|---|---|---|---|---|---|
| Logistic Regression | 55.44 | 51.06 | 60.66 | 0.9593 | 1.000 | 1642.8 |
| Random Forest | 88.50 | 84.01 | 93.50 | 0.9992 | 0.417 | 204.0 |
| XGBoost | **89.90** | **87.82** | 92.09 | **0.9998** | 0.989 | **8.5** |
| ECSEL | 79.08 | 68.10 | **94.27** | 0.9914 | 0.904 | 960.0 |

Table 20 compares ECSEL against standard baselines. XGBoost achieves the highest performance (89.90% F1, 87.82% recall), followed closely by Random Forest (88.50% F1, 84.01% recall). However, ECSEL achieves the highest precision (94.27%), indicating that when it flags a transaction as fraudulent, it is correct 94% of the time—a critical property for minimizing customer friction from false fraud alerts.

The optimal thresholds reveal different model behaviors: Random Forest operates at a relatively low threshold (0.417), while XGBoost and ECSEL require substantially higher confidence thresholds (0.989 and 0.904 respectively) to maintain precision-recall balance on this severely imbalanced dataset. Training times vary considerably: XGBoost is fastest at 8.5 seconds due to its highly optimized implementation, while ECSEL requires 16 minutes (960s), slower than tree-based methods but substantially faster than Logistic Regression (27 minutes). Moreover, ECSEL's inference cost is dominated by a single closed-form expression evaluation, making prediction orders of magnitude faster than ensemble methods that require traversing hundreds of decision trees.

Though ensemble methods achieve higher F1 scores, ECSEL provides full transparency through its explicit mathematical formula, enabling compliance verification and bias auditing while maintaining competitive performance.

**Comparison to Deep Symbolic Classification (DSC).** Visbeek et al. (2024) apply Deep Symbolic Classification to the same PaySim dataset, reporting F1 = 78.0%, recall = 67.0%, and precision = 95.0%. Their best-performing symbolic expression is given by

$$f = \sqrt{externalDest + CO} \cdot (Amount - MaxDest7 + T) \tag{A12}$$

with decision rule *isFraud* = 1 if $\sigma(f) > 0.7$, where $\sigma$ is the sigmoid function. Here, *externalDest* is a binary indicator for external recipient accounts, *CO* indicates cash_out transactions, *Amount* is the transaction amount, *MaxDest7* denotes the maximum transaction amount among the last seven transactions for the recipient account, and $T$ indicates TRANSFER transactions.

Our ECSEL implementation achieves slightly better performance with F1 = 79.08%, recall = 68.10%, and precision = 94.27%. However, important methodological differences complicate direct comparison.

DSC employs temporal aggregation features computed over the entire 30-day simulation period before train-test splitting. For example, *MaxDest7* represents the maximum transaction amount among the last seven transactions for a given recipient account, calculated using statistics from all transactions in the dataset. While the data was subsequently split 75/10/15 into

training, validation, and test sets, these aggregation features introduce two forms of data leakage: (1) temporal leakage, where features for a transaction at time $t$ incorporate information from future transactions at $t + \Delta t$, and (2) cross-set contamination, where training set features use statistics derived from transactions that ultimately appear in the test set. Although the authors acknowledge this limitation and mitigate it by adding Gaussian noise to aggregation features, the fundamental dependence on future transaction data remains.

In contrast, our feature set deliberately excludes such temporal aggregations to reflect production-realistic conditions where only historical transaction information is available at decision time. This methodological choice prioritizes deployment validity: in real-world fraud detection systems, decisions must be made in real-time using only past data, without access to statistics computed over future transactions.

Additionally, while exact training times are not reported, the authors mention DSC's computational requirement as a significant limitation that necessitated a single fixed train-validation-test split rather than cross-validation. Our gradient-based approach enables more efficient training (16 minutes on the full dataset) and supports 5-fold cross-validation for robust hyperparameter selection. Our slightly improved F1 score (79.08% vs. 78.0%) under production-realistic constraints suggests that ECSEL achieves competitive performance while maintaining stricter methodological validity and computational efficiency.

