# OpenReview forum: "ECSEL: Explainable Classification via Signomial Equation Learning"
_ICML.cc/2026/Conference — ICML 2026 regular_

### Official Review · Reviewer_D2w7 · 2026-03-11

**Soundness:** 3
**Presentation:** 3
**Significance:** 3
**Originality:** 3
**Overall Recommendation:** 5
**Confidence:** 3

**Summary:**

The paper proposes using signomial functions as an accurate and explainable replacement for standard classification functions. Generalizing polynomial functions, a signomial function is given by:
$$
f(x_1, \ldots, x_m) = \sum_{k=1}^K \alpha_{k} \prod_{j=1}^m x_j^{\beta_{k,j}}.
$$
where $K$ is a hyperparameter and the coefficients $\alpha_k$ and $\beta_{k,j}$ are learned. Since $\beta_{k,j}$ is a real number, the function can represent inverse and squareroot functions.

Contributions:
* The paper establishes that signomials are universal approximators.
* They describe how to train them, just using gradient approaches with an $\ell_1$ regularization term to encourage sparsity.
* Because signomial functions are simply linear combinations of product terms with real exponents, they satisfy several nice interpretability properties related to global feature attribution, counterfactual reasoning, gradient-based sensitivity, and more.
* The experiments compare the accuracy of signomial functions compared to random forests and XGBoost, finding that it is quite comparable. In addition, they do a case study showing how the resulting models can be interpreted.

**Compliance With Llm Reviewing Policy:**

Affirmed.

**Final Justification:**

The authors thoroughly responded to my points, and conducted an additional experimental evaluation.

I was concerned reading Reviewer dXcm's points, but I have been assauged by the authors' replies to Reviewer dXcm.

**Key Questions For Authors:**

1. Can you please describe the three most important properties and why we should care about them?

2. I would like to understand your contribution vs prior work. Could you please make this delineation more clear? For example, if I hadn't searched signomial functions, I might think you invented them since you don't cite the Duffin and Peterson papers.

**Limitations:**

Yes.

**Strengths And Weaknesses:**

Several thoughts in random order:

1. **Clarification of Novelty:** It's hard to tell what is novel in this paper and what isn't. Signomial functions were proposed by Duffin and Peterson in the 60s/70s. I'd be surprised if no one showed they're universal approximators until now. Ditto for how to train them (i.e., just gradient descent with regularization).

2. **Naming:** There are too many silly names in ML. Why are you trying to add another with "ECSEL"? Please just call this signomial learning for classification.

3. **Interpretability of Signomial Functions:** The main contribution of the paper seems to be a careful analysis of signomial functions and all the properties that are easy to read off from them. Honestly, this section (with 7 properties) is overwhelming and just reads as a list. I think the paper could really benefit from highlighting specific properties and how they compare to standard approaches. For example, if you say "here are the Shapley values of signomial functions for the SHAP function", "here are the feature importances", "here are the X", that would be very nice.

4. **Writing:** I generally like the paper. But the introduction, especially the related work, are not very good. Please rewrite to convey a story. Tbh, I think an LLM with the prompt to narrativize it would do wonders here.

5. **Experiments:** I appreciate the experiments on accuracy/F1/ recall. It would be great to see the interpretability in action e.g., here's the time to compute the Shapley values of XGBoost (via Tree SHAP), Signomial functions, random forests, etc. If you can compare and contrast the process of interpreting various functions that would be great.

---

> ### Author Rebuttal · Authors · 2026-03-31
>
> We thank the reviewer for the direct and constructive feedback. We understand the key points to be: (1) the novelty of the paper is hard to identify; (2) the seven interpretability properties are overwhelming and read as a list without grounding against standard methods; (3) a direct comparison of interpretability and computational cost against SHAP, LIME, and other methods is missing; and (4) minor points on the name and writing. We address these below.
>
> **(1) Novelty.** We thank the reviewer for raising this important point, and appreciate the directness. We will add citations to the foundational work of Duffin and Peterson in the revised manuscript, this is an oversight we are happy to correct.
>
> Regarding the universal approximation result: there is substantial prior work on signomials, most notably in geometric programming (Duffin & Peterson, 1973), but that field is concerned with minimizing signomial objectives, which is a different problem from ours (minimizing prediction error against observed data). To the best of our knowledge, after extensive literature search, the universal approximation property for signomials on the positive orthant has not been formally established before, and we provide it as our own contribution. To clarify the novelty more broadly: the contribution is not signomials themselves, nor gradient descent with regularization in isolation. Rather, it is the combination of (1) using signomials specifically as explainable classifiers competitive with standard baselines, (2) formalizing the interpretability properties that arise from this functional form, and (3) demonstrating that this targeted structure substantially outperforms state-of-the-art symbolic regression methods on signomial equations. We will make this delineation between prior work and our contributions much clearer in the revised manuscript.
>
> **(2) and (3) Interpretability properties and experiments.** These two points are closely related and we address them together. We have conducted a **dedicated interpretability experiment**, which we will include in the revised manuscript. Concretely, the experiment includes: (i) global feature attribution comparisons between ECSEL's G1 and TreeSHAP (RF, XGBoost), KernelSHAP (MLP), and LinearSHAP (LR); (ii) a direct G3 vs. LIME comparison across 200 test instances; (iii) a computational cost comparison across all methods; and (iv) a LIME seed stability analysis. Due to limited space, key findings and discussion are presented in the **response to Reviewer 1**; a summary table of explanation methods and computation times is provided below. Full results tables, instance-level comparisons, and an explicit mapping of each ECSEL property to its standard counterpart (G1 $\leftrightarrow$ global SHAP, L1/L2 $\leftrightarrow$ SHAP values, G3 $\leftrightarrow$ LIME) will be included in the revised manuscript, which we expect will also help the properties section read more concretely.
>
> **Summary of explanation methods, top-3 features, and total explanation time over the full test set. MLP (shallow) and MLP refer to architectures with 3 and 47 hidden units respectively.**
>
> | Method | Explainer | Explain time (s) | Top-3 features |
> |--------|-----------|-----------------|----------------|
> | ECSEL | Exact exponents | 0.1 | *PVER, ShopIntensity, PageValues* |
> | LR | Exact coefficients | 0.1 | *Month, IsReturning, Administrative* |
> | LR | LinearSHAP | 0.1 | *PVER, Month, ProductRelated* |
> | LR | LIME | 5.3 | *PVER, Month, ProductRelated* |
> | RF | TreeSHAP | 1.5 | *PVER, PageValues, ShopIntensity* |
> | RF | LIME | 32.0 | *PVER, PageValues, ExitRates* |
> | XGBoost | TreeSHAP | 0.1 | *PVER, Month, ShopIntensity* |
> | XGBoost | LIME | 7.7 | *PVER, ProductRelated, ExitRates* |
> | MLP (shallow) | KernelSHAP | 13.2 | *PVER, ProductRelated, Month* |
> | MLP (shallow) | LIME | 5.1 | *PVER, Administrative, ShopIntensity* |
> | MLP | KernelSHAP | 28.5 | *PVER, ProductRelated, Month* |
> | MLP | LIME | 5.7 | *PVER, ProductRelated, PageValues* |
>
> **(4) Last remarks.** On the name: we sympathize with this sentiment, however, the choice was partly practical as we plan to release ECSEL as a software package, and a distinct name aids discoverability. We also did not want to claim a general name such as "signomial classification," as others may develop different or improved approaches in the future. That said, we take the point seriously and will consider whether a less acronym-heavy presentation would serve the paper better. On writing: we will revise the introduction and literature review, number all equations, and add guided discussion of the toy example and figures, improving clarity and presentation throughout.
>
> ---
>
> Duffin, R.J. & Peterson, E.L. (1973). Geometric programming with signomials. *Journal of Optimization Theory and Applications*, 11, 3–35.

---

> > ### Author Rebuttal · Reviewer_D2w7 · 2026-04-01
> >
> > I thank the authors for their thorough reply to my points, as well as the additional experiments.
> >
> > I was concerned about the points raised by Reviewer dXcm. However, I am satisfied with the authors' response to Reviewer dXcm.

---

### Official Review · Reviewer_pDBE · 2026-03-13

**Soundness:** 4
**Presentation:** 3
**Significance:** 4
**Originality:** 3
**Overall Recommendation:** 5
**Confidence:** 5

**Summary:**

The authors propose a novel universal approximator, signomial functions. They develop a gradient-based optimization and regularization scheme to make these functions learnable and practically viable.They argue and demonstrate that this classification method is easily explainable and interpretable, as the learned coefficients are inherently understood by the explainee. They design certain interpretability and actionability properties that models should hold, and show that ECSEL models verify these properties. On symbolic regression and traditional classification benchmarks, ECSEL performs well while maintaining their transparency.

**Compliance With Llm Reviewing Policy:**

Affirmed.

**Final Justification:**

Fair comparison with MLPs has been done.

**Key Questions For Authors:**

1. How would a "deep signomial network" look like in practice? By that, I imply, how would the G1-3, D1-2 and L1-L2 change (or not) when adding more layers ?

2. You essentially argue that exponential coefficients are more easily understood than weighted sums... are you certain that this is an implicit claim you want to push through this paper ? If not, a small discussion on this distinction between "classical" neural networks and signomial networks would be beneficial.

3. For Fig3.d is it normal for attributions to be symmetric ? I find it evident that is mathematically so I'm not sure how it's relevant to highlight as an empirical finding ?

**Limitations:**

yes

**Strengths And Weaknesses:**

# Strengths

1. Truly innovative work that challenges assumptions in explainable AI and opens a new door for newly interpretable models.
2. Easy to follow, objectives are well posed and the paper answers the research questions rigorously
3. Clear how to expand the work to solve more problems.
4. Properties are clearly motivated and well defined.
5. Theorem 3.1 (Universal Approximation for Signomials) is vital and is provided, as well as the full proof (in appendix) and the proof sketch.
6. Strong empirical evidence that supports the claims made by the authors.

# Weaknesses

1.  As the authors demonstrate, a signomial function is mathematically equivalent to a two-layer neural network with non-linear activation function $\sigma = \exp$ (technically with bias as well).
$\to$ $K$ is equivalent the number of output neurons for a shallow network.
As such, I also _expect_ comparisons with shallow neural networks... (with the same basic architecture as the one proposed in the paper, i.e., a network per class, a final class neuron that takes the sum of the output neurons, softmax on all class neurons)
Further, the described network is much much more generalizable than what authors intended initially.
It is primordial to compare against networks for which
$$z_c(x)=\sum_{i=1}^K \left(w^{(c)}_i e^{b^{(c)}_i}\right) \sigma\left(W^{(c)}_i\cdot x\right)$$
$$\hat{z}(x) = \text{softmax}\left(z(x)\right),\quad \kappa(x) =\arg\max \hat{z}(x)$$

2. The submitted paper uses softmax scores as probabilities. No mention of calibration (or application to calibration).
3. Fig1 is a little small -- same for Fig2

---

> ### Author Rebuttal · Authors · 2026-03-31
>
> We thank the reviewer for the thoughtful and encouraging feedback, as well as the insightful questions regarding the relationship to neural networks, model capacity, and interpretability. We address the key points below.
>
> **(1) Equivalence to shallow neural networks.** We agree that comparing against a matched shallow architecture is valuable. We note that this point may concern either a conceptual or an empirical comparison, and we address both.
>
> While the architecture is related, ECSEL operates on log-transformed features ($\alpha_k \cdot \exp(\sum_j\beta_{kj}\log x_j)$) rather than raw features, and it is precisely this parameterization that enables the interpretability properties in Section 3.2. To empirically validate this comparison, we evaluated a shallow MLP baseline with 1 to 3 hidden units (matching ECSEL's $K=1,2,3$ terms) across all 11 datasets. The shallow MLP underperforms both ECSEL and standard MLPs (10 to 100 units) on most datasets: on 4 of 11 it yields 0% minority class recall (ILPD, TRANSFUSION) or poor accuracy (IRIS: 66.7%, CONTRACEPTIVE: 40.0%), while ECSEL achieves substantially higher F1 (e.g., ILPD: 74.39 vs. 59.69, CONTRACEPTIVE: 55.94 vs. 34.72). On simpler large-scale datasets (SKINNONSKIN, LOAN), both perform well. These results indicate that ECSEL's signomial structure makes more effective use of a limited parameter budget while additionally providing closed-form interpretability. Full results will be included in the revised appendix.
>
> **(2) Thoughts on "deep" signomial networks.** In this context, we interpret depth as increasing the number of signomial terms $K$, analogous to increasing capacity in shallow networks. All analytical properties (G1-G3, D1-D2, L1-L2) are derived for general $K$ and therefore hold without modification. For example, the G1 elasticity remains exact but becomes input-dependent through component weights rather than constant as in the $K=1$ case. In practice, increasing $K$ introduces a trade-off: while expressive power grows, interpretability becomes less immediate, as the equation becomes more complex to inspect. Empirically, we find that small $K \in [1,3]$ suffices for competitive performance, with larger $K$ offering diminishing returns. Identifying principled methods for selecting $K$ is an interesting direction for future work.
>
> **(3) Interpretability: signomial vs. neural network representations.** We agree that it is important to clarify the intended claim and thank the reviewer for pointing this out. We do not argue that exponential coefficients are inherently simpler than weighted sums. Rather, our point is that signomials provide a different form of interpretability: they encode multiplicative interactions and nonlinear scaling in a closed-form expression that can be directly read and reasoned about. For example, a coefficient $\beta_j$ indicates that a 1% increase in feature $j$ leads to a $\beta_j$% change in the output, which provides an intuitive elasticity-based interpretation.
>
> This perspective is consistent with the symbolic regression literature, where mathematical expressions of moderate complexity are considered interpretable because they can be inspected, verified, and analyzed (Rudin, 2019; Cranmer, 2023; Udrescu & Tegmark, 2020). We will clarify this distinction in the revised manuscript, including a discussion of how signomial networks relate to classical neural networks in both functional form and interpretability properties.
>
> **(4) Last remarks.** We thank the reviewer for the suggestions. We will (i) clarify the use of softmax outputs as probability-like scores and discuss calibration where appropriate, and (ii) improve the readability of Figures 1 and 2. Regarding the symmetry observed in Fig. 3(d), the reviewer is correct. In binary classification, symmetry is a mathematical consequence of the two class probabilities summing to one, making it redundant to show both classes. In the revised manuscript we will display only the positive class. We note that this symmetry is specific to the binary case (in multiclass settings it no longer holds, as attributions can distribute unevenly across classes).
>
> ---
>
> Rudin, C. (2019). Stop explaining black box machine learning models for high stakes decisions and use interpretable models instead. *Nature Machine Intelligence*, 1(5), 206–215.
>
> Cranmer, M. (2023). Interpretable machine learning for science with PySR and SymbolicRegression.jl. *arXiv preprint arXiv:2305.01582*.
>
> Udrescu, S.-M. & Tegmark, M. (2020). AI Feynman: a physics-inspired method for symbolic regression. *(NeurIPS 2020)*.

---

> > ### Author Rebuttal · Reviewer_pDBE · 2026-04-04
> >
> > The authors have answered my questions but I still disagree with the fair comparison with traditional MLPs.
> > The ECSEL network they train uses $K\times C$ "hidden neurons" and compare against a network with $K$ hidden neurons.
> >
> > To compare against mLPs, two comparisons would have needed to be done:
> > - each class gets a $K$ MLP for its logit, all scores are aggregated through a softmax
> > - same as above, but with log-transformed features.
> >
> > Authors should fairly compare their architecture against other architectures.
> >
> > Edit: the authors have correctly adressed the issue I raised, I will revert my score.

---

> > > ### Author Response · Authors · 2026-04-06
> > >
> > > We thank the reviewer for the clarification and apologize for the misunderstanding in our initial response. The reviewer correctly notes that ECSEL can be viewed as a per-class network with $\sigma = \exp$ on log-transformed features, and we appreciate the precise formulation provided. The requested comparison therefore isolates what this specific parameterization contributes relative to standard activations. We have now implemented both requested variants: $C$ separate $K$-neuron networks (one per class, $K \in [1,3]$), each producing a logit, aggregated via softmax, with (i) raw features and (ii) log-transformed features.
> > >
> > > We evaluated both across all 11 benchmark datasets using the same protocol as the main benchmark (Section 5.1): 5-fold stratified CV with Optuna TPE sampler, optimizing over $K\in[1,3]$, activation {$\in \text{relu, tanh}$}, learning rate, and batch size. The results can be found below.
> > >
> > > | Dataset | ECSEL Acc. | ECSEL F1 | ECSEL T(s) | PCM raw Acc. | PCM raw F1 | PCM raw T(s) | PCM log Acc. | PCM log F1 | PCM log T(s) |
> > > |---|---|---|---|---|---|---|---|---|---|
> > > | IRIS | **96.67** | **96.67** | 0.4 | **96.67** | **96.67** | 0.5 | 93.33 | 93.33 | 0.5 |
> > > | SEEDS | **97.62** | **97.62** | 0.1 | 92.86 | 92.77 | 0.8 | 88.10 | 87.31 | 0.7 |
> > > | HEARTS | **83.61** | **83.61** | 0.1 | 80.33 | 79.97 | 0.2 | **83.61** | 83.44 | 0.7 |
> > > | ILPD | **75.86** | **74.39** | 0.3 | 58.62 | 60.33 | 0.5 | 58.62 | 58.62 | 1.6 |
> > > | TRANSFUSION | **79.33** | 77.95 | 1.4 | 78.00 | **78.63** | 1.4 | 76.67 | 78.04 | 1.6 |
> > > | CONTRACEPTIVE | **56.27** | **55.94** | 4.4 | 53.22 | 53.57 | 1.2 | 53.90 | 54.73 | 0.4 |
> > > | COMPAS | **68.47** | **68.36** | 4.8 | 67.33 | 67.32 | 1.2 | 68.04 | 67.94 | 0.8 |
> > > | DEFAULT | **81.74** | **79.34** | 56.8 | 22.12 | 8.01 | 2.5 | 74.97 | 76.14 | 13.2 |
> > > | SKINNONSKIN | 99.25 | 99.25 | 108.5 | 99.86 | 99.86 | 118.1 | **99.88** | **99.88** | 147.7 |
> > > | MAMMOGRAPHY | **98.66** | **98.57** | 10.1 | 94.14 | 95.65 | 7.6 | 95.71 | 96.61 | 11.6 |
> > > | LOAN | **99.22** | **99.21** | 102.6 | 98.78 | 98.79 | 93.2 | 99.12 | 99.12 | 291.1 |
> > >
> > > PCM denotes the per-class MLP, and T the time in seconds. ECSEL achieves higher accuracy. and F1 on 10 of 11 datasets. Training times are comparable across methods, with no consistent advantage: ECSEL is faster on some datasets (e.g., SEEDS, HEARTS) and slower on others (e.g., LOAN). On 2 datasets (ILPD, DEFAULT), the per-class MLP performs substantially below ECSEL (and the other baselines), with DEFAULT showing a near-complete failure (22% accuracy, 8.01 F1 for the raw variant). A possible explanation could be that with ReLU activations and only 1-3 neurons per class, neurons with persistently negative pre-activations output zero and stop receiving gradient updates. ECSEL's signomial terms can produce very small or very large values, but each term $\alpha_k\prod_{j}x_j^{\beta_{k,j}}$ always produces a non-zero output for positive inputs, ensuring that gradients flow through every term during training. We note that Optuna selected ReLU over tanh on these datasets, despite both being available. Overall, the two architectures achieve broadly comparable performance, with ECSEL tending to be more robust across datasets while additionally providing closed-form interpretability through its signomial structure.
> > >
> > > To illustrate the interpretability difference, consider $K=2$ with 4 features. The per-class MLP computes $$z_0=v_1 \sigma(w_{11}x_1+\cdots+b_1)+v_2\sigma(w_{21}x_1+\cdots+b_2) + b_0,$$ where the activations introduce input-dependent gating: each neuron's weights are only active when the neuron fires, making feature effects region-dependent. For $K=1$ this reduces to a gated linear model (comparable to logistic regression in the active region), but for $K{\geq}2$ the neurons activate in different (or overlapping) regions of the input space, making the resulting feature weights region-dependent and difficult to summarize globally. ECSEL computes $$z_0=\alpha_1 x_1^{\beta_{1,1}} x_2^{\beta_{1,2}} x_3^{\beta_{1,3}} x_4^{\beta_{1,4}}+\alpha_2 x_1^{\beta_{2,1}} x_2^{\beta_{2,2}} x_3^{\beta_{2,3}} x_4^{\beta_{2,4}},$$ where each $\beta_{k,j}$ is the elasticity of term $k$ with respect to feature $j$: a 1% increase in $x_j$ changes the term's contribution by aggregation of $\beta_{k,j}$% values, and this holds globally across the input space. This structure directly yields closed-form elasticities (G1), exact counterfactuals (G2), and additive log-space decompositions (L1) from Section 3.2, none of which follow from a piecewise-linear parameterization.
> > >
> > > In practice, explaining the per-class MLP requires additional computation. Raw gradients (one backward pass) provide local sensitivity but not additive attribution. Global importance requires aggregation over the dataset, and proper additive decompositions require methods such as Integrated Gradients or SHAP. ECSEL provides all three directly from the learned expression. Full results will be included in the revised appendix.

---

### Official Review · Reviewer_dXcm · 2026-03-13

**Soundness:** 2
**Presentation:** 2
**Significance:** 3
**Originality:** 2
**Overall Recommendation:** 4
**Confidence:** 3

**Summary:**

The paper presents an ML approach called Explainable Classification via Signomial Equation Learning (ECSEL). The authors claimed it is an explainable ML approach, that can be used as alternative to traditional black-box ML models. The experiments are conducted using several datasets.

**Compliance With Llm Reviewing Policy:**

Affirmed.

**Final Justification:**

The authors provided detailed clarifications and additional results. Their response addressed my concerns. Therefore, I have raised my score accordingly.

**Key Questions For Authors:**

Please see the comments

**Limitations:**

yes

**Strengths And Weaknesses:**

The paper presents an ML approach called Explainable Classification via Signomial Equation Learning (ECSEL). The main concerns are as follows:


1- Based on Table 8 , (Full classification benchmark results across all datasets and metrics. pages 23-25), the predictive performance based on the evaluation measures (Accuracy, F1, Precision, Recall ),  shows that your approach outperformed the other baselines only on 4 datasets  out of 11 reported in the table. These results suggest that the existing popular baselines are better than your approach in terms of classification performance on 7 datasets. I am wondering what is the main reason behind considering your approach better?

2- In terms of training time, based on table 8, your approach has higher computation times on 8 datasets out of 11, while it is ranked as the second highest computation time on the remaining 3 datasets. What is the advantage of ECSEL in terms of computational time?

3- You used 5 fold cross validation, I suggest adding the standard deviation for each score in the tables (results).

4-  In Table 8, please report SVM and MLP performance using the LOAN (binary) dataset.

5- As you used 11 datasets, inculding 8 datasets for binary classification and 3 for multi-classification. If we focus only on 3 multi-class datasets namely (CONTRACEPTIVE (multi),SEEDS (multi), IRIS (multi)) , your approach outperformed the other baselines only on a dataset (SEEDS (multi)). This brings into question the limits of generalizability of this technique based on its performance on multi-classification tasks.

6- (Table 11, page 29), Test performance comparison of classification models on the Online Shopping Intention dataset. XGBoost and RF provided better predictive performance than  your approach  in terms of accuracy and F1 score. Also, they have reduced training time. In your opinion, what is the main advantage that leads one to use your technique not the other existing ones.

7- Your claimed that your approach ECSEL is an explainable method,  but there is not enough proof that supports the claim. To prove the advantage of your technique in terms of explainability, I suggest using the right evaluation measures, and existing explainable techniques as baselines to compare the quality of their explanations against your approach.

---

> ### Author Rebuttal · Authors · 2026-03-31
>
> We thank the reviewer for the careful and constructive feedback and we appreciate the opportunity to clarify the nature of ECSEL's contribution. We understand the main concerns to be: (1) ECSEL's predictive performance relative to strong baselines across datasets, including multi-class settings; (2) its computational cost during training, specifically on the Online Shopping Intent case study; (3) the need for a more rigorous validation of ECSEL's explainability, and (4) completeness and clarity of experimental reporting and small remaining points.
>
> **(1) Predictive performance.** We would like to clarify that ECSEL is not designed as a purely performance-driven method, but as an inherently interpretable alternative that remains competitive with state-of-the-art black-box models. It learns a closed-form signomial equation that is directly inspectable and supports exact feature attributions and sensitivity analysis without post-hoc approximation.
>
> Across the benchmark (Table 8), ECSEL achieves the highest F1 score on 4 of 11 datasets and ranks within one percentage point of the best-performing method on 9 of 11 datasets. Importantly, **no single baseline dominates across all datasets** (e.g., XGBoost, SVM, and RF each perform best on different subsets). The key distinction is that ECSEL additionally produces a closed-form, auditable model. In line with Rudin (2019), when interpretability is required, a transparent model with near state-of-the-art accuracy can be preferable to a marginally more accurate black-box model. We will make this positioning clearer in the revised manuscript.
>
> **(2) Computational cost.** ECSEL indeed has higher training times than several baselines. However, training efficiency is not the primary claim, and the current implementation uses standard optimizers (Adam and L-BFGS-B) without specialized engineering, unlike highly optimized methods such as XGBoost, which have undergone extensive algorithmic and systems-level engineering (e.g., optimized tree construction, parallelization, and hardware-aware implementations) over multiple releases. Despite this, training times remain practical. More importantly, ECSEL shifts cost from *post-hoc explanation* to *training*: once trained, all explanations are obtained in closed form at negligible cost. In contrast, black-box models require additional procedures such as SHAP or LIME, adding nontrivial computational overhead. We will make this distinction explicit and quantify explanation-time in the revision. Improving training efficiency is also a natural direction for future work, and recent optimizers such as Muon may further reduce fit time.
>
> For multi-class settings, ECSEL outperforms baselines on SEEDS, matches most methods on IRIS, and trails on CONTRACEPTIVE. As with the full benchmark, no single method dominates across all multi-class datasets. ECSEL remains consistently competitive while uniquely providing interpretable closed-form expressions per class. We agree that expanding evaluation on additional multi-class datasets would strengthen the claims, and we will do so in the revision.
>
> On the Online Shopping Intention dataset, while XGBoost and RF achieve slightly higher accuracy and F1 scores with lower training time, ECSEL provides a fundamentally different advantage: it yields a single closed-form equation that explicitly captures feature interactions (e.g., superlinear and inverse effects). This structure enables analytical capabilities such as exact sensitivity analysis, counterfactual reasoning, and additive decompositions, all derived directly from the model. We also note that ECSEL achieves the highest minority class recall (75.7%), which is particularly relevant in this imbalanced setting.
>
> **(3) Claim on explainability.** Closed-form expressions, as commonly studied under the field of symbolic regression, are widely recognized as inherently interpretable (Rudin, 2019; Cranmer, 2023; Udrescu & Tegmark, 2020), and we view this interpretability as a foundation for explainability. ECSEL builds on this by providing additional analytical properties (Section 3.2) that enable structured reasoning about predictions, such as exact sensitivities and local attributions. That said, we appreciate the suggestion and agree that direct comparison with established explanation methods could strengthen the paper. This point was raised by multiple reviewers as well, and so we conducted a **dedicated comparison with established methods** (TreeSHAP, KernelSHAP, LIME). Please see our full **response to Reviewer 1**.
>
> **(4) Last remarks.** Finally, we agree with the suggestions regarding experimental reporting. We will add standard deviations across all 5-fold CV results. We will also clarify the omission of SVM and MLP results on the LOAN dataset (due to prohibitive training time on this large-scale dataset), and either include these results or explicitly document the limitation.
>
> ---
> References are listed at the end of the response to Reviewer 3.

---

> > ### Author Rebuttal · Reviewer_dXcm · 2026-04-03
> >
> > I thank the authors for their detailed responses. I have raised my score accordingly.

---

### Official Review · Reviewer_4wCd · 2026-03-16

**Soundness:** 4
**Presentation:** 3
**Significance:** 4
**Originality:** 4
**Overall Recommendation:** 5
**Confidence:** 4

**Summary:**

The paper introduces or promote the use of Signomial Equations for classification. The relevance of this approach to the classic tasks is well introduced and several aspects regarding the benefits in terms of explainability are discussed. The evaluation on both synthetic and real world data is exhaustive.

**Compliance With Llm Reviewing Policy:**

Affirmed.

**Key Questions For Authors:**

> This bridges the gap between the interpretability of SR and the practical demands of modern classification tasks, offering a principled path toward explainable AI in high-stakes applications.

> Empirically, 45 out of 100 equations in the AI Feynman benchmark are signomials.

The first quote could be watered down given the second one.

> The signomial structure enables direct computation of global feature importance through gradient-based attribution.

This is not really studied in the main text, isn't it? Somehow this is the aspect of explainability missing.

> PROPERTY G3 (GRADIENT- BASED SENSITIVITY ).

This paragraph calls for a comparison with LIME

Also, it is good practice to number every equation.

> PROPERTY L2 (GRADIENT- BASED LOCAL ATTRIBUTIONS ).

How does L2 relate to G1? G1 says that for K>0, the explanation becomes local.

> Figure 3. Interpretability properties illustrated on a toy cancer screening example

Figure 3 is briefly mentioned in the text, but I think the explanations would benefit from guiding the reader through it.

**Limitations:**

The limitations are well discussed. Maybe what is missing is how SR behave in cases where the underlying model is not an SR.

**Strengths And Weaknesses:**

+ The paper is well written an easy to follow. Every arguments are well defended and back by clear and accessible theory.

- Although explainability is the first keyword, it remain with the impression that the aspect was not enough discussed through guided explanations. See below

- Comparisons with feature attribution methods is missing.

~ The papers is really dense and would certainly fit better in a journal where all aspects could be discussed.

---

> ### Author Rebuttal · Authors · 2026-03-31
>
> We thank the reviewer for the thoughtful and encouraging feedback. We appreciate the opportunity to address the key points below. We understand the key points to be: (1) a comparison with standard feature attribution methods is missing; (2) Property G1 is not sufficiently demonstrated in the main text; (3) a direct comparison of property G3 vs LIME is requested; (4) the relationship between properties L2 and G1 is unclear; (5) ECSEL's behavior when the underlying model is not signomial.
>
> To address points (1), (2), (3), we have conducted a **dedicated interpretability experiment**, a comparison desired by other reviewers as well. We computed feature attributions for ECSEL via its G1 property, alongside TreeSHAP for Random Forest (RF) and XGBoost, kernelSHAP on MLPs, and LinearSHAP on Logistic Regression (LR), and LIME for all methods on the Online Shopping Intent case study. Due to limited space, a summary of explanation methods and compute times is provided in the **response to Reviewer 4**; all remaining results mentioned below will be included as full tables in the revised paper.
>
> We found that all nonlinear methods identify *PageValue_per_ExitRate* (*PVER*) as the dominant predictor, while LR assigns highest importance to *Month*, which may reflect its limited capacity to capture multiplicative feature interactions. ECSEL produces these attributions at **zero computational cost** vs up to 28.5s for KernelSHAP. ECSEL's rankings align closely with tree-based methods ($\rho=0.87$ with RF-SHAP, $\rho=0.80$ with XGB-SHAP). Correlations are weaker for MLPs ($\rho=0.23$-$0.47$), though all nonlinear methods agree on the top predictor.
>
> For point (3), G3 gives exact, closed-form local sensitivity through the log-gradient, whereas LIME fits a local surrogate via sampling. To make this concrete, we computed both G3 and LIME attributions for 200 test instances using the same ECSEL model on the OSI case study. On a correctly predicted purchase instance ($p=0.746$), both identify *PVER*, *ShopIntensity*, and *PageValues* as the top features driving the prediction, with *ExitRates* and *Administrative* working against it. On a no-purchase instance ($p=0.524$), the same ranking and signs hold but with smaller magnitudes, reflecting the weaker signal near the decision boundary. Across all 200 instances, the mean Spearman rank correlation is $\rho=0.997$, with 100% sign agreement on all major features, while G3 is **~11,000x faster**. We note that this strong agreement may hold because LIME's surrogate is applied directly to ECSEL, potentially mirroring its structure in the local region. The choice of surrogate matters: when LIME is applied to a black-box model whose structure differs from a signomial, agreement with G3 naturally weakens, as we observe for MLPs in point (1).
>
> Beyond speed, LIME's sampling procedure introduces seed-dependent variance that G3 avoids entirely. LIME instability under random seeds is a well-documented limitation. Tan et al. (2023) show that standard LIME requires exponentially more samples than necessary for convergence, resulting in explanations that vary across seeds. Our experiments corroborate this: on a near-boundary instance, LIME rankings for MLP yield a mean pairwise Spearman $\rho$ of only 0.785 across 5 seeds, with individual seed correlations against ECSEL's G1 ranging from 0.46 to 0.66 (of which two statistically non-significant). ECSEL avoids this instability by construction, as its G3 attributions are derived analytically and require no sampling.
>
> Considering point (4), the reviewer correctly identifies the connection. For $K=1$, G1 yields a constant elasticity $E_{c,j}=\beta_{c,j}$ since the component weight reduces to 1, and L1 provides an exact additive decomposition in log-space. For $K>1$, the elasticity becomes input-dependent: $E_{c,j}(x)=\sum_k\frac{z_{ck}(x)}{z_c(x)}\beta_{ckj}$, a weighted average of per-term exponents. L2 builds on this via a first-order Taylor expansion in log-space around a reference point $x^\*$: $\phi_j=G_{c,j}(x^\*)(\log x_j-\log x_j^\*)$, turning G1's local sensitivity into an additive feature attribution relative to a baseline.
>
> For point (5), we conducted additional experiments fitting signomials to data generated from 8 non-signomial equations (rational, trigonometric, logarithmic) on $[1,5]^m$ for $K \in \{3,5,10\}$. For smooth, low-oscillation functions, ECSEL achieves $R^2>0.998$ even at small $K$, with $R^2$ improving monotonically, consistent with our universal approximation result. For strongly oscillatory functions, approximation quality remains poor regardless of $K$. We agree that 8 equations is too few for general conclusions and will expand the experiment and discussion in the final paper. We will also clarify the toy example, Figure 3, number all equations, and soften the SR-classification claim to better match our contribution.
>
> ---
>
> Tan, Z., Tian, Y., & Li, J. GLIME: general, stable and local LIME explanation. *(NeurIPS 2023)*

---

> > ### Author Rebuttal · Reviewer_4wCd · 2026-04-03
> >
> > The authors answered my concerns and questions.
> > I keep my score
> >
> > Thank you.
> >
> >
> > Just a comment. Next time refer to the reviewrs using their IDs, not number. I do not necessarily see them in the same order than you .

---

### Decision · Program_Chairs · 2026-04-30

**Decision:**

Accept (regular)

**Comment:**

This paper proposes ECSEL, a classification approach based on signomial equations, and argues that it offers a strong balance between competitive predictive performance and intrinsic interpretability.

The reviewers were overall positive. The paper was seen as technically solid, clearly written, and genuinely interesting in that it advances an interpretable model class rather than relying on post-hoc explanations. Several reviewers highlighted the formal analysis of interpretability properties as a major strength, and the empirical results were viewed as strong enough to support the paper's central claims.

The rebuttal further strengthened the submission. In particular, the authors addressed concerns about novelty and positioning relative to prior work on signomials, added the requested comparisons to matched shallow networks and to standard explanation methods such as SHAP and LIME, and clarified the intended scope of the interpretability claims. These additions appear to have resolved the main reviewer concerns.

The paper introduces a promising and well-supported direction for inherently interpretable classification, and I expect it to be of broad interest to researchers working on interpretable machine learning and symbolic modeling.